# REASONMAP: FINE-GRAINED ANNOTATION OF MATHEMATICAL DEFINITION AND THEOREMS IN LONG COT REASONING

## ABSTRACT

Annotating mathematical knowledge is a fundamental prerequisite for structured knowledge acquisition from large-scale mathematical solutions. However, existing datasets lack the fine-grained annotations of theorems and definitions at scale. This makes it challenging to assess how well modern mathematical reasoning models understand and apply specific knowledge within their complex chain-of-thought (CoT) outputs. In this paper, we propose a new task: Automatic Annotation of Mathematical Definitions and Theorems (AAMDT), which aims to extract Mathematical Definitions and Theorems (MDTs) from long CoT reasoning. To tackle this, we introduce ReasonMap, a novel two-stage training framework. The first stage, Foundational Model Training, builds a broadly capable model by performing Supervised Fine-Tuning on a hybrid corpus of both concise human annotations and LLM-augmented long CoT data. The second stage, High-Fidelity Alignment, then refines this model using Direct Preference Optimization to ensure the final output is both precise and reliable. Comprehensive experiments show ReasonMap consistently outperforms strong baselines on the AAMDT task, especially in long CoT scenarios. Crucially, we validate the quality of our annotations by demonstrating that the extracted MDTs can directly enhance the mathematical reasoning performance of downstream large language models. Our work offers a scalable solution for the automatic annotation of mathematical corpora, significantly reducing the reliance on manual labeling. The github repository can be found at: `https://anonymous.4open.science/r/ReasonMap-6A83`.

## 1 INTRODUCTION

Enhancing the mathematical reasoning of large language models (LLMs) is a central challenge in modern AI, where progress is fundamentally tied to the quality of training data (OpenAI et al., 2024; DeepSeek-AI et al., 2025a). Primary efforts to improve data quality focus on increasing coarse-grained diversity, such as training on different problem categories (e.g., "Algebra") (He et al., 2025). While beneficial for generalization, such coarse-grained labels provide limited insight into the model's actual reasoning process. To facilitate a more granular analysis of model behavior, a fine-grained annotation becomes an urgent necessity, which motivates our work on annotating theorems and definitions used in reasoning chains.

Current annotation approaches fall into two main paradigms: human annotation and automatic annotation. Human annotation by experts produces high-quality datasets such as GSM8K (Cobbe et al., 2021) and Natural-Proof-gen (Welleck et al., 2022), but this process is prohibitively costly to scale. More recently, automatic annotation offers an alternative for creating large datasets like DeepMath-103K (He et al., 2025). However, it remains financially demanding due to its reliance on API-based LLMs and typically generates coarse-grained labels (see Figure 1.(a)). Crucially, both annotation approaches struggle with annotating long chain-of-thought (CoT) solutions, which are essential for promoting LLMs' test-time performance on hard problems (Snell et al., 2025). As shown in Figure 1.(b), when the difficulty of problems escalates, the number of definitions and theorems that a problem may cover will dramatically increase, making it extremely challenging to annotate correctly and efficiently for complex mathematical solutions at scale.

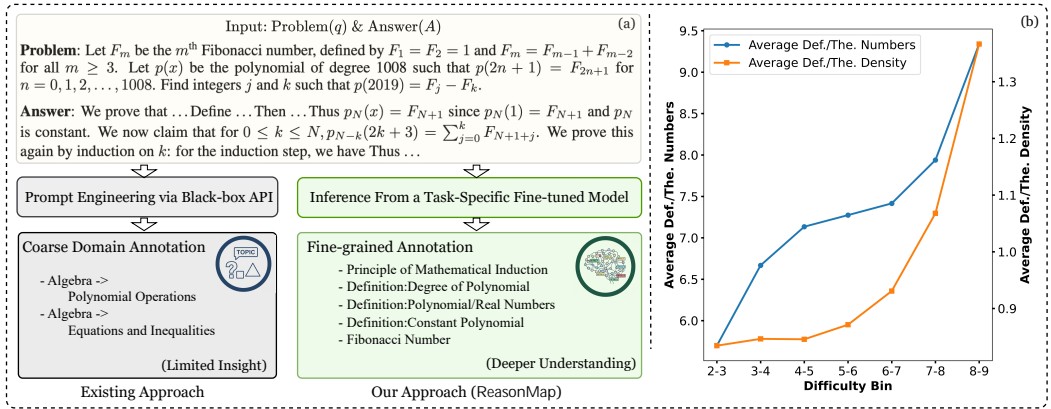

Figure 1: (a) A comparison of annotation granularity. **Coarse-grained**: High-level domain tags (e.g., "Algebra"). **Fine-grained**: specific, Atomic knowledge units (e.g., "Principle of Mathematical Induction") (b) In DeepMath-103K, the average number and density of required definitions/theorems increase with problem difficulty. Details are in Appendix A.3.

In order to address the shortcomings of existing methods, we propose the task of Automatic Annotation of Mathematical Definitions and Theorems (AAMDT). Unlike previous approaches that assign coarse-grained labels, AAMDT aims to extract the fine-grained theorems and definitions embedded within reasoning steps. Solving this task poses two fundamental challenges: 1) How to train a model to efficiently capture precise, fine-grained mathematical concepts? 2) How to generalize the learned policy to current knowledge-intensive scenarios like long context in complex CoT reasoning solutions?

To this end, we propose ReasonMap, a novel two-stage framework. The core of our framework is **Stage 1: Foundational Model Training**, which focuses on knowledge acquisition and long-context generalization. We train the model on human-annotated solutions as a knowledge warmup, followed by an augmented dataset with "short-to-long" CoTs to enable its generalization to complex reasoning scenarios. This training process equips our model with domain knowledge and long-context robustness for the AAMDT task. Subsequently, to further enhance the model's precision, we introduce **Stage 2: High-Fidelity Alignment**. In this optimization phase, we employ Direct Preference Optimization (DPO) (Rafailov et al., 2023) on a generated dataset of pairwise preferences to refine the model's output quality and mitigate subtle flaws. This staged approach allows us to build a scalable and robust annotator, which demonstrates a significant performance improvement over strong baselines on the AAMDT task. The practical utility of this annotator is ultimately validated by its ability to improve downstream mathematical reasoning tasks. We summarize the key contributions as follows:

- We introduce and formalize AAMDT task, addressing the critical need for scalable and cost-effective fine-grained labels in mathematical corpora.

- We develop ReasonMap, a novel two-stage framework to tackle AAMDT that enables high-fidelity annotation of long CoT reasoning at scale.

- Our evaluation demonstrates that ReasonMap significantly outperforms advanced baselines, and we validate the practical utility of its annotations by showing they can directly improve the reasoning of downstream LLMs.

## 2 RELATED WORKS

### 2.1 THE IMPACT OF DATA DIVERSITY ON LARGE LANGUAGE MODELS

Data diversity has emerged as a critical factor in improving LLM performance and generality during post-training stages such as SFT, DPO and reinforcement learning (Yang et al., 2025a; Wen et al., 2025; Yang et al., 2025b). From the perspective of training dynamics, Jin et al. (2025) highlight that diverse data is essential to prevent entropy collapse and avoid convergence to suboptimal local min-

ima during RL. In mathematical reasoning, ReasonFlux (Yang et al., 2025a) provides evidence that leveraging diverse compositional reasoning templates to generate varied CoTs markedly strengthens the reasoning ability of LLMs. To ensure comprehensive coverage across mathematical domains, Chen et al. (2025) proposed a knowledge-system-driven synthesis framework that constructs problem sets based on structured mathematical concepts, thereby guaranteeing topic diversity and systematically enhancing model generalization. Large-scale datasets such as DeepMath-103K (He et al., 2025) further support robust generalization by covering a broad spectrum of mathematical domains and problem types. Collectively, these efforts highlight that increasing the diversity of mathematical data, especially through the targeted selection of various problem types, can significantly enhance the accuracy and generalization of LLMs in mathematical tasks.

## 2.2 ANNOTATION OF MATHEMATICAL REASONING DATASETS

A core strategy for enhancing LLM's mathematical reasoning ability involves constructing high-quality datasets for model training. Existing annotation approaches fall into two paradigms.

**Human annotation**. Datasets like GSM8K (Cobbe et al., 2021) and MATH (Hendrycks et al., 2021) cover elementary to university-level problems, with MATH incorporating topic and difficulty labels from AoPS[1]. Subsequent efforts, including SAT (Azerbayev et al., 2024) and OCWCourses (Lewkowycz et al., 2022), have sought to increase the difficulty of the problem by curating more advanced questions from existing curricular resources. Natural-Proofs-Gen (Welleck et al., 2022) uniquely offers fine-grained annotations of theorems and definitions involved in each proof step. However, human annotation remains labor intensive and costly to scale, limiting its applicability to large corpora or long CoT data.

**LLM-based automatic annotation**. Recent work has explored the use of LLMs to automate both solution generation and labeling. For example, the Open-R1 dataset[2] employs LLMs to generate and validate solution paths, assigning coarse-grained category tags. Omni-Math (Gao et al., 2024) introduces hierarchical domain labeling based on a predefined taxonomy of mathematical topics, though its difficulty annotation still relies on AoPS–derived human standards. DeepMath-103K (He et al., 2025) further estimates difficulty by calculating pass rates, a process that requires making numerous API calls for each problem to gather statistics. Still, they remain computationally expensive and typically provide only high-level annotations (e.g., Algebra), with limited insight into the specific mathematical knowledge used. Despite growing interest in large-scale annotation, existing techniques fall short in providing definition-level and theorem-level granularity, which is essential for deeper model evaluation and knowledge-centered data curation.

## 3 METHOD

### 3.1 TASK FORMULATION

We formulate the Automatic Annotation of Mathematical Definitions and Theorems (AAMDT) task as a conditional sequence generation problem. The goal is to train a model that, given a mathematical text as input, autoregressively generates a structured text output enumerating the relevant Mathematical Definitions and Theorems (MDTs).

Formally, the model learns a policy $\pi$ that approximates the probability distribution $P(Y|X)$. The input $X \in \mathcal{Q} \cup \mathcal{C}_{\text{def}}$ is a text sequence constructed from either:

- A problem-answer pair $(q, A) \in \mathcal{Q}$, containing a math problem $q$ and its corresponding reasoning chain $A$.
- A definition-only instance $c \in \mathcal{C}_{\text{def}}$, representing a conceptual explanation of a math definition.

The output $Y$ is the target sequence of MDTs, $L = \{d_1, \ldots, d_G, \ t_1, \ldots, t_M\}$, where $d_g$ and $t_m$ represent the $g$-th definition and $m$-th theorem, respectively.

This formulation rests on two core hypotheses regarding mathematical reasoning:

---

[1] https://artofproblemsolving.com/
[2] https://huggingface.co/datasets/open-r1/OpenR1-Math-220k

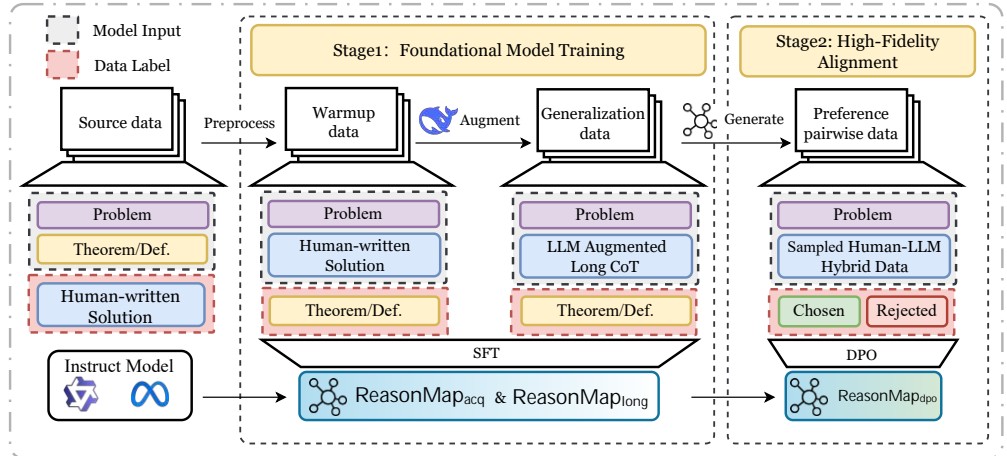

Figure 2: An overview of our two-stage framework ReasonMap. Stage 1 focuses on Foundational Model Training. We first perform SFT on the Warmup data to obtain an initial model, ReasonMap$_{acq}$. This model is then sequentially trained on the Generalization data, which contains LLM Augmented Long CoT, to produce the foundational model, ReasonMap$_{long}$. Stage 2 is for High-Fidelity Alignment. Here, the ReasonMap$_{long}$ is used to generate outputs, from which we construct Preference pairwise data containing Chosen and Rejected examples. Finally, we apply DPO to obtain the optimized model, ReasonMap$_{dpo}$.

- **Decomposability:** Motivated by prior research (Cho et al., 2025; Zhang et al., 2025), we assume that a complex mathematical reasoning process can be decomposed into a series of discrete knowledge steps. This structure is evident in datasets like NaturalProofs-Gen (Welleck et al., 2022), where proofs are explicitly constructed as sequences of steps linked to specific theorems or definitions.

- **Path-Knowledge Alignment:** While prior work demonstrated that conditioning on MDTs help generate valid proofs (Welleck et al., 2022) (Knowledge → Path), we posit that the inverse is also learnable. That is, a valid CoT reasoning path implicitly aligns with a unique set of supporting MDTs. Our task addresses this inverse problem by learning the mapping from the reasoning path back to its foundational knowledge (Path → Knowledge).

This formulation allows AAMDT to serve as a systematic approach for analyzing reasoning models. By annotating unstructured solutions with a structured layer of MDTs, we enable: 1) fine-grained interpretability of how LLMs apply specific knowledge, moving beyond surface-level correctness; and 2) the extraction of MDTs as contextual hints to enhance the reasoning performance of downstream models.

## 3.2 THE FRAMEWORK OF REASONMAP

As illustrated in Figure 2, our framework follows a two-stage design philosophy: we first build a knowledgeable model via foundational training, and then refine its output for precision and reliability through high-fidelity alignment. The following sections detail each stage.

### 3.2.1 STAGE 1: FOUNDATIONAL MODEL TRAINING

The core of Stage 1 is to holistically address both knowledge acquisition and generalization. We recognized that simply training on one type of data is insufficient. Human-written answers, while they are gold standard for accuracy, are often too concise. LLM-generated CoTs, while providing the necessary length and complexity for generalization, may lack the precision of human expertise. Therefore, we developed a hybrid training corpus that leverages the strengths of both, creating a comprehensive learning experience for the model. This corpus has two key components: a warmup dataset for knowledge acquisition and a generalization dataset for short-to-long CoT adaptation.

First, we source dataset from Natural-Prover-Gen (Welleck et al., 2022), a high-quality dataset of human-written mathematical proofs. Each entry in this source data provides a math problem $q$, a

human-written answer $A$, and corresponding ground-truth MDTs $L$ from Proofwiki[3]. To suit the specific needs of our AAMDT task, we process and reconstruct these entries to create our warmup dataset $\mathcal{D}_{\text{acq}}$ (examples can be found in Appendix A.9), aiming to provide the model with a robust conceptual foundation. It comprises two complementary data types:

$$\mathcal{D}_{\text{acq}} = \underbrace{\{(q, A, L) \mid (q, A) \in \mathcal{Q}\}}_{\text{problem-answer}} \cup \underbrace{\{(s, L) \mid s \in \mathcal{S}_{\text{def}}\}}_{\text{definition-only}}. \tag{1}$$

Next, to mitigate overfitting and ensure the model's robustness, we construct the generalization dataset, designed specifically for short-to-long CoT adaptation. We begin by sampling problem-answer pairs from $(q, A, L)$ in $\mathcal{D}_{\text{acq}}$. For each instance, we use the concise human-written answer $A$ and corresponding problem $q$ to seed the generation of a more elaborate reasoning chain. Specifically, we prompt the DeepSeek-R1 (DeepSeek-AI et al., 2025b) to produce a long CoT answer $A_l$, following the instructions provided in Appendix A.2. This process results in a new training triple $(q, A_l, L)$ and yields an augmented training set $\mathcal{D}_{\text{long}}$, which is used in subsequent steps.

We utilize the two collected datasets for training and evaluation. First, we partition $\mathcal{D}_{\text{acq}}$ into training $\mathcal{D}_{\text{acq}}^{\text{train}}$ and test sets $\mathcal{D}_{\text{acq}}^{\text{test}}$. The generalization sets, $\mathcal{D}_{\text{long}}^{\text{train}}$ and $\mathcal{D}_{\text{long}}^{\text{test}}$, are curated by augmenting subsets sampled from $\mathcal{D}_{\text{acq}}^{\text{train}}$ and $\mathcal{D}_{\text{acq}}^{\text{test}}$, respectively, thereby preventing data leakage. Our training protocol is a two-step curriculum process. We first warm up the base model on $\mathcal{D}_{\text{acq}}^{\text{train}}$, yielding $\pi_{\text{acq}}$. Subsequently, we continue training this model on the generalization data $\mathcal{D}_{\text{long}}^{\text{train}}$ to adapt its knowledge to complex long CoT reasoning.

The final output of this stage is the trained foundational model $\pi_{\text{long}}$. By fine-tuning on both a knowledge acquisition data and long CoT generalization data, $\pi_{\text{long}}$ becomes proficient at accurately extracting MDTs from both concise human-written and long CoT scenarios.

### 3.2.2 STAGE 2: ACHIEVING HIGH-FIDELITY MDTs ALIGNMENT

Despite the effectiveness of SFT, $\pi_{\text{long}}$ still exhibits subtle hallucinations and format inconsistencies (see Appendix A.15). To achieve high-fidelity alignment, we employ DPO, selected for its superior training stability and computational efficiency compared to complex RLHF pipelines (Yang et al., 2025b). However, the critical challenge lies in constructing effective rejected responses against the ground-truth chosen ones. Motivated by Cheng et al., which constructs preference pairs from hard failure cases via minimal perturbations, we specifically target **hard negatives**—candidates generated by $\pi_{\text{long}}$ that are plausible but contain fine-grained flaws. This strategy forces the model to discern subtle nuances rather than obvious errors, a capability crucial for precise annotation and explicitly validated by our subsequent experiments.

We first construct a generation dataset $\mathcal{D}_{\text{gen}}$, which is composed of all samples from the $\mathcal{D}_{\text{long}}^{\text{train}}$ combined with a filtered subset of examples from the $\mathcal{D}_{\text{acq}}^{\text{train}}$. Note that samples from $\mathcal{D}_{\text{long}}^{\text{train}}$ make up a slightly larger proportion; the goal is to maintain a strong focus on long CoT capabilities while preserving the model's generalization to shorter cases. For the acquisition subset, we only include samples where the label length $|L| > 3$, considering that overly short labels are prone to being trivially copied by the model and fail to produce meaningful preference comparisons.

For each sample $(q, A) \in \mathcal{D}_{\text{gen}}$, we perform inference using the model $\pi_{\text{long}}$ to generate $N$ candidate outputs $\{\hat{L}_1, \hat{L}_2, \ldots, \hat{L}_N\}$. Here, $\hat{L}_k \leftarrow \pi_{\text{long}}(q, A)$ represents a complete sequence of MDTs generated in the $k$-th sampling trial. Each is compared with the ground-truth label $L$ using two metrics: **Match Rate** $m_k$: the proportion of items in $\hat{L}_k$ that exactly match items in the ground-truth label $L$; and **Repeat Rate** $r_k$: the proportion of repeated items in $\hat{L}_k$. The distribution for the Repeat Rate shows a sharp peak at 0, whereas the Match Rate is highly concentrated near 1.0 (see Figure 4 in Appendix A.5).

Consequently, we define a scoring function $\mathcal{S}$ to evaluate each candidate output $\hat{L}_k$ based on its match rate $m_k$ and repeat rate $r_k$:

$$\mathcal{S}(\hat{L}_k) = (1 - r_k) \cdot m_k. \tag{2}$$

---

[3] https://proofwiki.org

To select a challenging but flawed negative example, we identify the candidate with the highest score that is lower than a certain quality threshold $\gamma$. The index of this "rejected" candidate $k^*$ is thus determined by:

$$k^* = \arg\max_{k \in \{1,\ldots,N\}} \mathcal{S}(\hat{L}_k), \quad \text{s.t. } \mathcal{S}(\hat{L}_k) < \gamma. \tag{3}$$

With the selection index $k^*$ identified for each input, we proceed to construct our final preference dataset $\mathcal{D}_{\text{dpo}}$. For each preference tuple $(q, A, L^+, L^-) \in \mathcal{D}_{\text{dpo}}$, the chosen response $L^+$ is the corresponding ground-truth MDT label, while the rejected response $L^-$ is the candidate $\hat{L}_{k^*}$ selected by equation 3. We then adopt the DPO objective, which aligns the model's preferences by leveraging these pairwise comparisons between the Chosen $L^+$ and Rejected $L^-$ outputs, as follows:

$$\mathcal{L}_{\text{dpo}} = -\mathbb{E}_{\mathcal{D}_{\text{dpo}}} \left[ \log \sigma \left( \beta \log \frac{\pi_{\text{dpo}}(L^+ \mid (q, A))}{\pi_{\text{long}}(L^+ \mid (q, A))} - \beta \log \frac{\pi_{\text{dpo}}(L^- \mid (q, A))}{\pi_{\text{long}}(L^- \mid (q, A))} \right) \right]. \tag{4}$$

Here, $\beta > 0$ is a temperature parameter controlling the strength of the preference, while the sigmoid function $\sigma(\cdot)$ converts the preference score into a probability for the loss function. The reference model is an initial copy of $\pi_{\text{long}}$ whose parameters are kept frozen during training. The DPO alignment stage refines the foundational model $\pi_{\text{long}}$ from Stage 1 by correcting subtle flaws like format inconsistencies and hallucinations. The resulting model $\pi_{\text{dpo}}$ is not only capable of handling both short and long CoT reasoning but also demonstrates higher fidelity.

# 4 EXPERIMENTS

We conduct a comprehensive set of experiments to validate ReasonMap's effectiveness and utility. Our evaluation focuses on three core performance aspects: First, we assess its performance by comparing it against strong baselines on diverse in-distribution (ID) and out-of-distribution (OOD) test sets. Second, we demonstrate its methodological reliability via detailed ablation studies that justify our design choices. Third, we conduct a downstream problem-solving task to investigate whether the extracted MDTs can serve as effective contextual hints to directly enhance the mathematical reasoning capabilities of other LLMs.

Beyond technical efficacy, we also address the economic feasibility of our framework. We provide a comparative cost analysis in Appendix F, demonstrating that our approach reduces annotation costs by approximately 68% compared to commercial APIs on large-scale datasets. Detailed experimental settings are presented in Appendix A.4. Furthermore, to ensure the reliability of our synthetic training data, we include a quantitative quality analysis using ROUGE metrics (see Appendix E), confirming that our guided generation strategy induces substantial reasoning expansion rather than simple imitation.

## 4.1 STATISTICS OF THE GENERATED DATA SETS

The statistics of the datasets obtained through our processing pipeline are summarized in Table 4 in the Appendix A.5. For training, $\mathcal{D}_{\text{acq}}^{\text{train}}$ serves as the primary acquisition set with 39,885 entries, averaging 2.63 MDTs and 90.17 tokens per sample. To capture long CoT reasoning, $\mathcal{D}_{\text{long}}^{\text{train}}$ includes 5,802 entries with substantially more MDTs (6.30 on average) and much longer sequences (1,103.26 tokens). In addition, two DPO datasets, $\mathcal{D}_{\text{dpo-Llama}}$ and $\mathcal{D}_{\text{dpo-Qwen}}$, contain 7,637 and 6,655 entries respectively, with average lengths of 735.20 and 705.66 tokens.

For evaluation, $\mathcal{D}_{\text{acq}}^{\text{test}}$ are constructed: the first comprising 1,135 samples with moderate length (184.86 tokens), and $\mathcal{D}_{\text{long}}^{\text{test}}$ containing 562 much longer samples (1,190.47 tokens), both maintaining around 6 MDTs per instance. In addition, we also employ two external test sets: $\mathcal{D}_{\text{DeepMath}}^{\text{test}}$ with extremely long contexts (3,069.56 tokens on average) and $\mathcal{D}_{\text{Omni}}^{\text{test}}$ with mid-length sequences (365.32 tokens). Together, these evaluation datasets provide a comprehensive evaluation of the model's ID and OOD performance.

## 4.2 EVALUATION

**Baselines** Previous works mostly classify mathematical reasoning chains by prompting powerful LLMs like the ChatGPT series (Gao et al., 2024; He et al., 2025). We follow this precedent but

Table 1: Performance comparison of different models and training stages across multiple test sets. Metrics include precision (P), recall (R), F1 score, and accuracy (ACC).

| Models | In distribution (ID) | | | | | | Out-of distribution (OOD) | |
|---|---|---|---|---|---|---|---|---|
| | $\mathcal{D}_{\text{acq}}^{\text{test}}$ | | | $\mathcal{D}_{\text{long}}^{\text{test}}$ | | | $\mathcal{D}_{\text{DeepMath}}^{\text{test}}$ | $\mathcal{D}_{\text{Omni}}^{\text{test}}$ |
| | P | R | F1 | P | R | F1 | ACC | ACC |
| Frontier LLMs | | | | | | | | |
| GPT4o-mini | 0.692 | 0.723 | 0.707 | 0.458 | 0.632 | 0.531 | 0.427 | 0.466 |
| GPT5-mini | 0.632 | 0.708 | 0.667 | 0.505 | 0.609 | 0.552 | 0.460 | 0.476 |
| Deepseek-V3 | 0.756 | 0.717 | 0.735 | 0.676 | 0.682 | 0.678 | 0.420 | 0.411 |
| deepseek-V3.1 | 0.779 | 0.671 | 0.721 | 0.523 | 0.588 | 0.553 | 0.420 | 0.428 |
| Qwen2.5-7B-Instruct | | | | | | | | |
| Base-Model | 0.610 | 0.611 | 0.610 | 0.443 | 0.580 | 0.502 | 0.412 | 0.401 |
| ReasonMap-7B$_{\text{acq}}$ | **0.833** | **0.844** | **0.838** | 0.488 | **0.805** | 0.607 | 0.357 | 0.376 |
| ReasonMap-7B$_{\text{long}}$ | 0.804 | 0.798 | 0.800 | 0.716 | 0.766 | **0.740** | 0.434 | 0.350 |
| ReasonMap-7B$_{\text{dpo}}$ | 0.786 | 0.766 | 0.775 | **0.732** | 0.737 | 0.734 | **0.455** | **0.413** |
| Llama3.1-8B-Instruct | | | | | | | | |
| Base-Model | 0.577 | 0.649 | 0.611 | 0.359 | 0.702 | 0.475 | 0.321 | 0.365 |
| ReasonMap-8B$_{\text{acq}}$ | **0.788** | **0.801** | **0.794** | 0.432 | **0.791** | 0.558 | 0.316 | 0.342 |
| ReasonMap-8B$_{\text{long}}$ | **0.788** | 0.749 | 0.768 | 0.725 | 0.742 | 0.733 | 0.400 | **0.389** |
| ReasonMap-8B$_{\text{dpo}}$ | 0.758 | 0.742 | 0.749 | **0.789** | 0.715 | **0.750** | 0.432 | **0.389** |

establish our baselines based on two distinct selection criteria. First, to assess the task's inherent difficulty using cost-effective frontier models with strong instruction-following capabilities, we select GPT-4o-mini and DeepSeek-V3 (DeepSeek-AI et al., 2025b) and their next-generation models, GPT5-mini and DeepSeek-V3.1. Second, to isolate the effectiveness of our proposed framework, we evaluate the widely used dense models that serve as our backbone—Qwen2.5-7B-Instruct and Llama3.1-8B-Instruct. Using a unified prompt, we compared their performance with our method on the AAMDT task. The prompt for these baselines is presented in Appendix A.11.

**In-Distribution Evaluation.** We use $\mathcal{D}_{\text{acq}}^{\text{test}}$ and $\mathcal{D}_{\text{long}}^{\text{test}}$ as in-distribution (ID) test data. Specifically, models take in the ID test data and predict a sequence of MDTs. To measure predictions, we employ an LLM-as-Judge approach (Gu et al., 2024) where an LLM matches MDTs between labels and generated outputs. We calculated the precision (P), recall (R), and their F1 score (F1 = 2P×R/(P + R)) as evaluation metrics. The detailed calculation and prompt are presented in Appendix A.12.

**Out-of-Distribution Evaluation.** To assess the out-of-distribution (OOD) performance of ReasonMap, we additionally introduce two test sets for evaluation. We uniformly sampled 1,000 instances based on domain distributions from DeepMath-103K and Omni-Math datasets, denoted as $\mathcal{D}_{\text{DeepMath}}^{\text{test}}$ and $\mathcal{D}_{\text{Omni}}^{\text{test}}$, respectively. The OOD test sets only annotate broad domain labels (see Figure 1 (a)). It is infeasible to compute the exact matches directly. Therefore, we employ a proxy evaluation: we prompt an LLM judge to verify if the fine-grained MDTs generated by our model fall within the ground-truth coarse domain of the problem. This assesses the semantic relevance of the outputs in an OOD context; the detailed prompt is presented in Appendix A.13. We acknowledge that this coarse-grained proxy metric might theoretically inflate performance by rewarding trivial or generic concepts (e.g., "Definition: Set") that technically satisfy domain constraints but offer little reasoning value. To investigate this, we conducted a human evaluation on the MDTs classified by the LLM judge (detailed in Appendix D.3).

### 4.3 MAIN RESULTS

**ReasonMap substantially improves long CoT reasoning accuracy.** As shown in Table 1, compared to baseline LLMs, the trained models ReasonMap (-long and -dpo) achieve significantly better performance in both in-distribution (ID) and out-of-distribution (OOD) scenarios.

On $\mathcal{D}_{\text{long}}^{\text{test}}$, accuracy improves by 7–10% over the baselines. On the challenging $\mathcal{D}_{\text{DeepMath}}^{\text{test}}$ test set, we observe an additional 1–3% gain over most baselines. The notable exception is GPT-5-mini, which demonstrates leading performance in OOD generalization. However, this OOD advantage does not translate to In-Distribution (ID) evaluations; surprisingly, GPT-5-mini even exhibits a slight performance regression compared to its predecessor GPT-4o-mini on $\mathcal{D}_{\text{acq}}^{\text{test}}$ (F1 drops from 0.707 to 0.667), whereas our ReasonMap models maintain consistent superiority across both ID and OOD tasks. The results also demonstrate that our strategy effectively bridges the generalization gap. In particular, we observe that the precision and accuracy steadily increase as the training framework moves on. Beyond long CoT improvements, the models also generalize well to shorter reasoning tasks: on $\mathcal{D}_{\text{acq}}^{\text{test}}$, they consistently exceed the baselines. In particular, ReasonMap-7B$_{\text{dpo}}$ even matches DeepSeek-V3 on $\mathcal{D}_{\text{Omni}}^{\text{test}}$. We attribute these improvements to two key aspects:

- The comprehensive Stage 1 training establishes a robust knowledge anchor. By training on a hybrid dataset of both concise human-written solutions ("warmup") and detailed LLM-augmented long CoTs ("generalization"), this stage simultaneously equips the model with accurate foundational knowledge and the ability to generalize. As corroborated by our qualitative error analysis (Appendix D.2), this rigorous grounding effectively mitigates hallucinated reasoning and over-extraction, specifically reducing high-risk "Inference Drift" errors compared to baselines.

- The subsequent Stage 2 alignment with DPO provides critical refinement. This stage is crucial for achieving high-fidelity annotations. It addresses the subtle flaws remaining after the first stage, which is clearly reflected in the consistent increase in precision and the improved precision-recall balance (see Figure 3).

**Progressive training improves the balance between precision and recall.** A closer look at our baselines in Table 1 reveals the inherent difficulty of the AAMDT task for general Frontier LLMs. The API-based models, for instance, exhibit key weaknesses on $\mathcal{D}_{\text{long}}^{\text{train}}$. GPT-4o-mini, GPT-5-mini, and Deepseek-V3.1 show a significant gap between their precision and recall. This precision-recall imbalance is even more pronounced in the non-fine-tuned base models. While the Deepseek-V3 model achieves a better balance between precision (0.676) and recall (0.682), its overall F1 score of 0.678 remains suboptimal. These baseline results underscore a crucial point we discussed earlier: without task-specific priors from fine-tuning, even powerful models struggle with the precision required for this task.

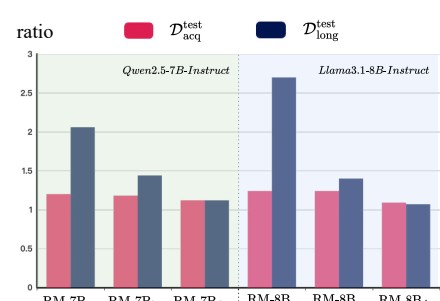

Figure 3: The change in the ratio between the number of predicted MDTs and the number of MDTs in the labels during training.

Another clear trend emerges when analyzing different training stages. Following Stage 1, the initial model $\pi_{\text{acq}}$ tends to over-generate MDTs in long CoT test sets (e.g., $\mathcal{D}_{\text{long}}^{\text{test}}$), leading to higher recall but lower precision. As training advances with $\mathcal{D}_{\text{long}}^{\text{train}}$, the gap between precision and recall narrows. In Stage 2, even precision occasionally surpassing recall. Figure 3 confirms this by showing that the ratio between the number of predicted MDTs and the number of MDTs in the labels steadily decreases toward 1, indicating an increase in precision and confidence in long CoT predictions.

**Qwen-based models generally outperform Llama-based one.** Across backbones, Qwen2.5-7B-Instruct models consistently deliver stronger results than Llama3.1-8B-Instruct in most evaluation settings. Although Llama-based models slightly surpass Qwen on $\mathcal{D}_{\text{long}}^{\text{test}}$ after DPO training, Qwen maintains a superior overall performance across the test sets.

## 4.4 ABLATION STUDY

**Is separating the two SFT steps a good choice?** To validate our choice of a sequential training protocol for Stage 1, we compare two approaches. The first is a *joint* training strategy, where the model is trained on a merged dataset containing both warmup $\mathcal{D}_{\text{acq}}^{\text{train}}$ and generalization $\mathcal{D}_{\text{long}}^{\text{train}}$ simultaneously. The second is the *sequential* strategy, where the model is first trained exclusively on the warmup data and then subsequently fine-tuned on the generalization data. The results, as

Table 2: Ablation study on different variants of training strategies.

| Training Method | In distribution (ID) | | | | | | Out-of distribution (OOD) | |
|---|---|---|---|---|---|---|---|---|
| | $\mathcal{D}_{\text{acq}}^{\text{test}}$ | | | $\mathcal{D}_{\text{long}}^{\text{test}}$ | | | $\mathcal{D}_{\text{DeepMath}}^{\text{test}}$ | $\mathcal{D}_{\text{Omni}}^{\text{test}}$ |
| | P | R | F1 | P | R | F1 | ACC | ACC |
| ReasonMap-7B$_{\text{long}}$ | | | | | | | | |
| - *sequential* | **0.804** | 0.798 | 0.800 | **0.716** | **0.766** | **0.740** | **0.434** | 0.350 |
| - *joint* | 0.780 | **0.830** | **0.804** | 0.653 | 0.701 | 0.676 | 0.410 | **0.416** |
| - w/o *hint* | 0.733 | 0.776 | 0.753 | 0.649 | 0.694 | 0.670 | 0.415 | 0.362 |
| - *sequential-ep*1 | 0.732 | 0.780 | 0.755 | 0.649 | 0.751 | 0.691 | 0.431 | 0.406 |
| - *drop*-1*k* | 0.758 | 0.816 | 0.785 | 0.618 | 0.772 | 0.685 | 0.419 | 0.402 |
| - *drop*-3*k* | 0.718 | 0.765 | 0.740 | 0.618 | 0.762 | 0.682 | 0.398 | 0.388 |
| ReasonMap-7B$_{\text{dpo}}$ | | | | | | | | |
| - $\gamma = 0.9$ | **0.786** | **0.766** | **0.775** | **0.732** | **0.737** | **0.734** | **0.455** | **0.413** |
| - $\gamma = 0.7$ | 0.784 | 0.762 | 0.772 | 0.724 | 0.715 | 0.719 | 0.451 | 0.397 |
| - $\gamma = 0.5$ | 0.781 | 0.740 | 0.759 | **0.732** | 0.728 | 0.729 | 0.448 | 0.401 |

shown in Table 2, clearly reveal that the sequential strategy outperforms the joint approach. This indicates that establishing a strong foundational knowledge base on high-quality human data first provides a better starting point before generalizing to the complexities of long CoT reasoning. To further validate this, we tested a variant *sequential-ep*1, in which we limited the training on the warmup data to a single epoch. The resulting performance drop confirms that a thorough warmup phase is crucial for the overall success of Stage 1.

**Is including definition-only samples beneficial for overall performance?** To justify our decision to include the definitions-only samples $\mathcal{C}_{\text{def}}$, we evaluate two experimental configurations: *drop*-1*k* (removing 1,000 definition-only samples) and *drop*-3*k* (removing 3,000 definition-only samples). Compared to the result of the complete training set, we observe consistent performance degradation in both *drop* variants. This empirically validates that maintaining a comprehensive collection of definition-only samples is crucial to establishing a solid knowledge base and ensuring high-precision concept extraction.

**Are hints necessary for aligning reasoning paths in training data annotation?** A core challenge in creating our generalization dataset is the existence of multiple valid solution paths for a single mathematical problem. This presents a critical data alignment issue: if the LLM generates a $A_l$ for one valid path, while our ground-truth MDTs were annotated for another, the resulting ($A_l$, $L$) pair becomes a misaligned and noisy sample. To demonstrate it, we reconstruct $\mathcal{D}_{\text{long}}^{\text{train}}$ without a guided answer $A$ (*w/o hint*), allowing the LLM to solve problems freely. Performance decline observed in both ID and OOD evaluations, consequently revealing that training on the unconstrained $A_l$ fundamentally degraded ReasonMap's learning process and its final accuracy.

**Can harder negative samples make DPO training more effective?** We conduct an ablation study on the hyperparameter $\gamma$, which serves as the quality threshold for selecting rejected candidates in the DPO dataset. We change the value of $\gamma$ (from 0.5 to 0.9) and reconstruct $\mathcal{D}_{\text{dpo-Qwen}}$. The evaluation results show that employing a higher $\gamma$ value yields better performance. This suggests that introducing harder negative samples improves the model's ability to distinguish subtle differences in MDTs quality.

## 4.5 ENHANCING MATHEMATICAL REASONING WITH EXTRACTED MDTs

To validate the quality and practical utility of the MDTs extracted by ReasonMap, we designed a mathematical problem-solving experiment. We evaluated two base models, Qwen2.5-7B-Instruct and Llama3.1-8B-Instruct, on two test sets: MATH500[4], a collection of moderately difficult problems, and $\mathcal{D}_{\text{DeepMath}}^{\text{test}}$, a subset of highly complex problems from DeepMath-103k. We compared model performance in two settings: a baseline condition using a direct

---

[4]https://huggingface.co/datasets/HuggingFaceH4/MATH-500

prompt, and an MDTs-augmented condition where relevant MDTs, extracted by ReasonMap from each problem's corresponding ground-truth solution, were injected into the prompt as contextual hints. The specific prompt structure is detailed in Appendix A.7. The results, presented in Table 3, demonstrate the consistent effectiveness of our approach. Across all models and datasets, augmenting prompts with MDTs led to improved accuracy. Critically, the performance gains are most pronounced on the more challenging $\mathcal{D}_{\text{DeepMath}}^{\text{test}}$, where accuracy increased by 1.2% for Qwen2.5-7B-Instruct and a significant 2.0% for Llama3.1-8B-Instruct. This strongly suggests that for complex problems nearing the limits of a model's intrinsic knowledge, MDTs provide crucial and directional guidance that helps construct a correct reasoning chain.

Conversely, the smaller gains on the simpler MATH500 dataset indicate a marginal effect; the models already possess a high baseline capability, thus the added knowledge provides diminishing returns. A case study in Appendix A.6 further illustrates how MDTs guide a model toward a more rigorous and correct line of reasoning.

Table 3: The accuracy of models with MDTs and without MDTs on the MATH500 and $\mathcal{D}_{\text{DeepMath}}^{\text{test}}$ datasets.

| Method | MATH500 | $\mathcal{D}_{\text{DeepMath}}^{\text{test}}$ |
|---|---|---|
| Qwen2.5 Model | 0.740 | 0.506 |
| + MDTs | 0.759 | 0.518 |
| Llama3.1 Model | 0.483 | 0.302 |
| + MDTs | 0.497 | 0.322 |

Beyond these setting, we further explored the robustness and scalability of our method through two additional experiments: 1) By extracting MDTs from the model's own rollouts rather than ground truth, we observed consistent performance gains (e.g., +1.1% on MATH500 for Qwen2.5), validating the practical utility of ReasonMap in inference-only scenarios (see Appendix C for details). 2) To verify effectiveness on stronger reasoning models, we evaluated DeepSeek-V3 on the highly challenging AIMO benchmark[5]. Remarkably, explicit MDT injection yielded a **5.56%** accuracy boost (see Appendix B.2), demonstrating that even advanced models benefit significantly from precise knowledge grounding.

## 5 CONCLUSION

In this paper, we present the novel task of Automatic Annotation of Mathematical Definitions and Theorems (AAMDT), designed for the systematic extraction of fine-grained Mathematical Definitions and Theorems (MDTs). Addressing the lack of such annotations in existing datasets, we developed ReasonMap, a novel two-stage framework. The first stage, Foundational Model Training, employs SFT on a hybrid corpus of human annotations and LLM-augmented long CoT data to build a knowledgeable and generalizable model. The second stage, High-Fidelity Alignment, then uses Direct DPO to refine the model's outputs, ensuring high precision and reliability. Our evaluation on both in-distribution (ID) and out-of-distribution (OOD) settings demonstrates that ReasonMap significantly outperforms strong baselines in the accurate extraction of MDTs, particularly in challenging long CoT reasonings. To further validate the quality of these annotations, we show that the extracted MDTs can directly enhance the mathematical reasoning performance of other LLMs. This result not only confirms the efficacy of our approach but also underscores the value of fine-grained knowledge for building more capable and verifiable reasoning systems.

---

[5] https://huggingface.co/datasets/AI-MO/aimo-validation-aime

## 6 ETHICS STATEMENT

This work adheres to standard ethical guidelines for academic research. All datasets used for training and evaluation in this study were constructed from publicly available sources, ensuring that no private or sensitive data was involved. The base large language models we utilized for fine-tuning are also publicly accessible. The research focuses on the annotation of mathematical knowledge, a task with no foreseeable negative societal impacts or ethical concerns. We are committed to transparency and responsible research practices.

## 7 REPRODUCIBILITY STATEMENT

We are committed to ensuring the reproducibility of our research. We have provided a detailed description of all experimental settings, hyperparameters, and the computational platform in the Appendix A.4. Upon publication, we will release all the training data constructed for our SFT and DPO stages, the evaluation datasets, and the source code for our data processing pipeline and evaluation scripts. Furthermore, the final trained model weights for our ReasonMap models will also be made publicly available.

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

# A APPENDIX

## A.1 USE OF LLMS

In the preparation of this manuscript, we utilized a large language model for assistance. The model's role was strictly limited to language editing, such as improving grammar, clarity, and phrasing of the text. All core contributions–including the formulation of the research problem, the design of the methodology and experiments, the execution of the experiments, and the analysis and interpretation of the results–were conducted entirely by the authors. The authors take full responsibility for the final content and all claims made in this paper.

## A.2 GENERATION PROMPT

---

**Generation Prompt**

You are an advanced reasoning assistant tasked with generating a mathematical proof for a given problem. I will provide you with a human-written, concise proof outline. Your task is to generate a complete proof from scratch, following the logical structure and key steps of the provided outline, but without explicitly referencing or acknowledging the human proof. Your proof should: 1) Adhere to the reasoning style of a logical, step-by-step mathematical proof. 2) Incorporate all relevant mathematical concepts and principles implied by the problem. 3) Avoid any direct mention of the human proof; treat it as if you are deriving the proof entirely independently. 4) Maintain clarity and rigor, ensuring that each step is justified and flows naturally from the previous.

---

## A.3 APPLICATION TO A LARGE-SCALE MATHEMATICAL DATASET

To further demonstrate the practical importance of ReasonMap, we perform a case study using ReasonMap$_{dpo}$ to extract MDTs from each sample in the DeepMath-103K dataset. For each difficulty level, we computed the average number of extracted MDTs $N_{it}$. We also calculated the average item density for each difficulty, defined as:

$$\text{Density} = \frac{N_{it}}{\sqrt{\text{steps}}}, \tag{5}$$

where "steps" are determined by splitting the solution at each occurrence of "\n\n" (Yang et al., 2025c). Note that we omitted the easiest and hardest difficulty groups due to insufficient data in these categories. As shown in Figure 1 (b), our analysis reveals that the average number and density of MDTs are higher in more difficult problems.

## A.4 EXPERIMENTAL SETTING

The DPO dataset was constructed by generating $N=4$ responses per prompt and applying a filtering threshold of $\gamma$ of 0.9. In the evaluation process, we selected GPT-4o-mini as our judge. Recent work has demonstrated its strong capabilities to handle complex tasks (Gu et al., 2024; Yang et al., 2025b; Teng et al., 2025) and its efficiency in terms of faster response and output times, making it a highly suitable choice for robust and scalable evaluation. Crucially, to validate its reliability for our ID and OOD annotation task, we conducted a human evaluation (detailed in Appendix D.1). The results demonstrated a high agreement rate ($> 90\%$) between GPT-4o-mini and human experts, confirming it as a highly suitable choice for robust and scalable evaluation.

We selected Qwen2.5-7B-Instruct and Llama3-8B-Instruct as the backbone models for ReasonMap, and conducted full-parameter training on our three-stage dataset. To further validate the adaptability of our framework to next-generation architectures, we also conducted experiments using the more recent Qwen3-8B as a backbone (results detailed in Appendix B). All experiments were performed on an Ascend Atlas 800T A2 server equipped with $8 \times 64$GB NPUs, utilizing the MindSpeed-LLM training framework for model optimization and acceleration. In the first stage of training, we set the learning rate to 1e-5, the batch size to 32, and the maximum sequence length to 1k tokens. Training was performed for 3 epochs using a cosine decay learning rate schedule, with a warmup ratio of 0.01. During the second stage, the learning rate was adjusted to 5e-6, and the maximum sequence length was increased to 8k tokens, while all other hyperparameters remained unchanged. For the DPO training stage, we set $\beta$ to 0.1 and used the sigmoid function as the reward transformation. To mitigate reward hacking (Skalse et al., 2022) and stabilize the training, we incorporated SFT loss into the DPO training, with a loss coefficient of 0.5. The learning rate was set to 5e-7, and training was conducted for one epoch. We use the same system prompt across all stages, as shown in Appendix A.14.

A.5 SUPPLEMENTARY TABLE AND FIGURE

| Dataset | Total | Avg MDTs | Avg len |
|---|---|---|---|
| $\mathcal{D}_{\text{acq}}^{\text{train}}$ | 39,885 | 2.63 | 90.17 |
| $\mathcal{D}_{\text{long}}^{\text{train}}$ | 5,802 | 6.30 | 1,103.26 |
| $\mathcal{D}_{\text{dpo-Llama}}$ | 7,637 | – | 735.20 |
| $\mathcal{D}_{\text{dpo-Qwen}}$ | 6,655 | – | 705.66 |
| $\mathcal{D}_{\text{long}}^{\text{test}}$ | 1,135 | 6.17 | 184.86 |
| $\mathcal{D}_{\text{long}}^{\text{test}}$ | 562 | 6.29 | 1190.47 |
| $\mathcal{D}_{\text{DeepMath}}^{\text{test}}$ | 1,000 | – | 3069.56 |
| $\mathcal{D}_{\text{Omni}}^{\text{test}}$ | 1,000 | – | 365.32 |

Table 4: Statistics of different datasets.

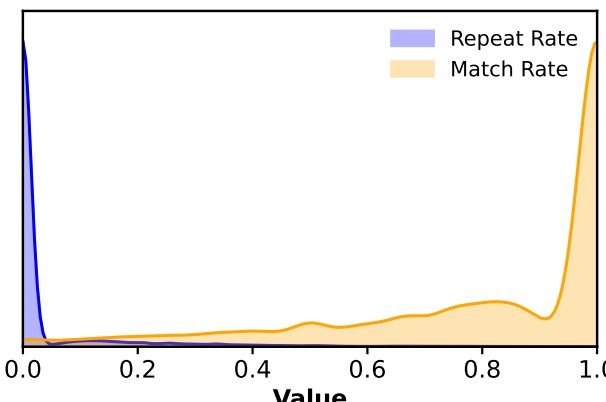

Figure 4: The distribution of repeat rate and exact match rate in $\mathcal{D}_{\text{gen}}$.

## A.6 CASE STUDY

### Question

Evaluate the contour integral $\int_{\gamma} \tanh(z)\, dz$ using the residue theorem, where $\gamma(t) = e^{it}$ for $0 \leq t \leq 2\pi$.

**Relevant theorems and concepts:**

- Residue Theorem
- Poles of Hyperbolic Tangent Function
- Definition: Order of Pole/Simple Pole
- Definition: Isolated Singularity
- Definition: Analytic Function/Complex Plane
- Contour Integral of Hyperbolic Tangent Function

### Wrong Answer (w/o MDTs)

To evaluate the contour integral $\int_{\gamma} \tanh(z)\, dz$ where $\gamma(t) = e^{it}$ for $0 \leq t \leq 2\pi$, we will use the residue theorem. The residue theorem states that for a function $f(z)$ that is analytic inside and on a simple closed contour $\gamma$ except for a finite number of singular points inside $\gamma$, the integral is given by

$$\int_{\gamma} f(z)\, dz = 2\pi i \sum \text{Res}(f, a_i)$$

where the sum is over all the singular points $a_i$ inside $\gamma$.

First, we identify the singular points of the function $\tanh(z)$. The hyperbolic tangent function $\tanh(z) = \frac{\sinh(z)}{\cosh(z)}$ has singular points where $\cosh(z) = 0$. The function $\cosh(z) = \frac{e^z + e^{-z}}{2}$ has zeros at $z = i\left(\frac{\pi}{2} + k\pi\right)$ for any integer $k$. The only singular points of $\tanh(z)$ inside the unit circle $|z| = 1$ are $z = i\frac{\pi}{2}$. Next, we need to find the residue of $\tanh(z)$ at $z = i\frac{\pi}{2}$. The residue of a function $f(z)$ at a simple pole $z = a$ is given by

$$\text{Res}(f, a) = \lim_{z \to a}(z - a)f(z)$$

For $\tanh(z)$ at $z = i\frac{\pi}{2}$,

$$\text{Res}\left(\tanh(z), i\frac{\pi}{2}\right) = \lim_{z \to i\frac{\pi}{2}}\left(z - i\frac{\pi}{2}\right)\tanh(z) = \lim_{z \to i\frac{\pi}{2}}\left(z - i\frac{\pi}{2}\right)\frac{\sinh(z)}{\cosh(z)} = \lim_{z \to i\frac{\pi}{2}}\frac{\sinh(z)}{\cosh'(z)}$$

Since $\cosh'(z) = \sinh(z)$, we have

$$\text{Res}\left(\tanh(z), i\frac{\pi}{2}\right) = \lim_{z \to i\frac{\pi}{2}}\frac{\sinh(z)}{\sinh(z)} = 1$$

Thus, the residue of $\tanh(z)$ at $z = i\frac{\pi}{2}$ is 1. By the residue theorem, the value of the integral is

$$\int_{\gamma} \tanh(z)\, dz = 2\pi i \cdot 1 = 2\pi i$$

Therefore, the answer is

$$\boxed{2\pi i}$$

**Correct Answer (w MDTs)**

To evaluate the contour integral $\int_\gamma \tanh(z)\, dz$ where $\gamma(t) = e^{it}$ for $0 \leq t \leq 2\pi$, we will use the residue theorem. The residue theorem states that for a function $f(z)$ that is analytic inside and on a simple closed contour $\gamma$ except for a finite number of singular points inside $\gamma$, the integral of $f(z)$ around $\gamma$ is given by

$$\int_\gamma f(z)\, dz = 2\pi i \sum_k \text{Res}(f, a_k)$$

where the sum is over all the singular points $a_k$ inside $\gamma$.

First, we identify the singular points of the function $\tanh(z)$. The hyperbolic tangent function $\tanh(z) = \frac{\sinh(z)}{\cosh(z)}$ has singularities where $\cosh(z) = 0$. The solutions to $\cosh(z) = 0$ are $z = \frac{(2n+1)\pi i}{2}$ for $n \in \mathbb{Z}$. These are the points where $\cosh(z)$ has simple zeros.

Next, we need to determine which of these singular points lie inside the contour $\gamma(t) = e^{it}$ for $0 \leq t \leq 2\pi$. The contour $\gamma$ is the unit circle in the complex plane, so we need to check if any of the points $z = \frac{(2n+1)\pi i}{2}$ lie within the unit disk $|z| < 1$.

For $n = 0$, we have $z = \frac{\pi i}{2}$, and $|\frac{\pi i}{2}| = \frac{\pi}{2} > 1$. For $n = -1$, we have $z = -\frac{\pi i}{2}$, and $|-\frac{\pi i}{2}| = \frac{\pi}{2} > 1$.

For $n \neq 0, -1$, the magnitudes $|\frac{(2n+1)\pi i}{2}| = \frac{|2n+1|\pi}{2} > 1$.

Thus, there are no singular points of $\tanh(z)$ inside the unit disk. Since the function $\tanh(z)$ is analytic inside and on the unit circle, the integral of $\tanh(z)$ over the unit circle is zero.

Therefore, the value of the contour integral is

$$\boxed{0}$$

Guided by the theorem, the model strictly adhered to all its application conditions. It correctly calculated all the poles of $\tanh(z)$ as $z = \frac{(2n+1)\pi i}{2}$. Following this, it performed the most critical logical step: verifying whether any of these poles were located inside the path of integration (the unit circle). It correctly found that $|\frac{\pi i}{2}| \approx 1.57 > 1$, and therefore, no poles reside within the contour. According to the Residue Theorem, the integral is correctly determined to be zero.

Although also knowing how to use the Residue Theorem and identifying the poles, a model without relevant theorems and concepts has a fatal logical error at this crucial juncture. It incorrectly asserted that the pole $z = \frac{i\pi}{2}$ is located inside the unit circle. This logical fallacy sent it down a completely erroneous computational path, prompting it to solve for a residue that should not have been considered, which ultimately yielded an incorrect, non-zero answer.

### A.7 System prompt for Problem-solving

**Prompt with MDTs**

Your task is to solve the Math problem. You will be provided with relevant theorems and concepts as a reference to assist you. Please use this information to guide your reasoning and solution. If applicable, include the final answer in \boxed{} for closed-form results like multiple choices or mathematical solutions.
{question}
Relevant theorems and concepts:
{MDTs}

**Prompt without MDTs**

Your task is to solve the Math problem. Please use this information to guide your reasoning and solution. If applicable, include the final answer in \boxed{} for closed-form results like multiple choices or mathematical solutions.

A.8 INSTANCES IN NATURAL-PROOFS-GEN

---

**A definition-only instance in Natural-Proofs-Gen**

```
<prompt>
<definition>
<title> Definition:Finite Group/Axioms </title>
<content>
A [[Definition:Finite Group|finite group]] is an
[[Definition:Algebraic Structure|algebraic structure]]
$\struct {G, \circ}$ which satisfies the following four conditions:
{{begin-axiom}}
{{axiom | n = \text{FG} 0
        | lc= [[Definition:Closed Algebraic Structure|Closure]]
        | q = \forall a, b \in G
        | m = a \circ b \in G
}}
{{axiom | n = \text{FG} 1
        | lc= [[Definition:Associative Operation|Associativity]]
        | q = \forall a, b, c \in G
        | m = a \circ \paren{b \circ c} = \paren{a \circ b} \circ c
}}
{{axiom | n = \text{FG} 2
        | lc= [[Definition:Finite Set|Finiteness]]
        | q = \exists n \in \N
        | m = \order G = n
}}
{{axiom | n = \text{FG} 3
        | lc= [[Definition:Cancellable Operation|Cancellability]]
        | q = \forall a, b, c \in G
        | m = c \circ a = c \circ b \implies a = b
}}
{{axiom | m = a \circ c = b \circ c \implies a = b
}}
{{end-axiom}}
These four stipulations are called the finite group axioms.
</content>
</definition>
```

---

**A problem-answer instance in Natural-Proofs-Gen**

```
<prompt>
<theorem>
<title> Real Number Subtracted from Itself leaves Zero </title>
<content>
Let $x \in \R$ be a real number. Then:$x - x = 0$
where $x - x$ denotes the operation of real subtraction.
</content>
</theorem>
<reference> Inverses for Real Addition </reference>
<proof>
{{begin-eqn}}
{{eqn | l = x - x
     | r = x + \paren{-x}
     | c = {{Defof|Real Subtraction}}
}}
{{eqn | r = 0
     | c = [[Inverses for Real Addition]]
}}
{{end-eqn}}
{{qed}}
</proof>
```

## A.9 AN EXAMPLE OF $\mathcal{D}_{\text{ACQ}}^{\text{TRAIN}}$

---

**An example of $\mathcal{D}_{\text{acq}}^{\text{train}}$**

**Question** ($q$)

Let $G$ be a cyclic group. Let $g$ be a generator of $G$. Let $\phi$ be an automorphism on $G$. Then $\phi(g)$ is also a generator of $G$.

**Answer** ($A$)

By definition of automorphism, $\phi$ is a homomorphism. It follows that this result is a specific instance of Homomorphic Image of Cyclic Group is Cyclic Group. $\square$

**MDTs** ($L$)

- Homomorphic Image of Cyclic Group is Cyclic Group
- Definition:Group Automorphism
- Automorphism Maps Generator to Generator
- Definition:Cyclic Group/Generator
- Definition:Group homomorphism

---

## A.10 AN EXAMPLE OF $\mathcal{D}_{\text{LONG}}^{\text{TRAIN}}$

---

**An example of $\mathcal{D}_{\text{long}}^{\text{train}}$**

**Question ($q$)**

Let $G$ be a cyclic group. Let $g$ be a generator of $G$. Let $\phi$ be an automorphism on $G$. Then $\phi(g)$ is also a generator of $G$.

**Long CoT Answer ($A_l$)**

Okay, so I need to prove that if $G$ is a cyclic group generated by an element $g$, and $\phi$ is an automorphism of $G$, then the image of $g$ under $\phi$, which is $\phi(g)$, is also a generator of $G$. Let me start by recalling some basic definitions and properties here.

First, a cyclic group is a group that can be generated by a single element. That means every element in $G$ can be written as some power of $g$, right? So $G = \{g^0, g^1, g^2, \ldots, g^{n-1}\}$ if $G$ is finite of order $n$, or all integer powers if it's infinite cyclic.

An automorphism $\phi$ is a bijective homomorphism from $G$ to itself. Since $\phi$ is a homomorphism, it preserves the group operation. So for any integers $k$ and $m$, $\phi(g^k) = \phi(g)^k$, because homomorphisms respect exponents. Also, being bijective means $\phi$ is both injective and surjective.

Now, the goal is to show that $\phi(g)$ generates $G$. Since $G$ is cyclic and generated by $g$, every element in $G$ is of the form $g^k$ for some integer $k$. Because $\phi$ is surjective, every element in $G$ must be the image of some element under $\phi$. But since $\phi$ is a homomorphism, the image of any element in $G$ under $\phi$ can be expressed as $\phi(g)$ raised to some power. Wait, let me think again.

Since $\phi$ is an automorphism, it's surjective, so every element in $G$ must be hit by $\phi$. But $\phi$ is a homomorphism, so for any $h$ in $G$, there exists some $a$ in $G$ such that $\phi(a) = h$. But since $G$ is cyclic, $a = g^m$ for some $m$. Therefore, $h = \phi(g^m) = \phi(g)^m$. That means every element $h$ in $G$ is some power of $\phi(g)$, which would imply that $\phi(g)$ generates $G$. Therefore, $\phi(g)$ is a generator.

Alternatively, maybe I should approach this by considering that the image of a cyclic group under a homomorphism is cyclic. Since $\phi$ is surjective (because it's an automorphism), the image of $G$ under $\phi$ is $G$ itself. Then, since $G$ is cyclic, the image must be generated by the image of the generator $g$, which is $\phi(g)$. Therefore, $\phi(g)$ must generate $G$.

Wait, but the homomorphic image of a cyclic group is cyclic, yes. Since $\phi$ is a homomorphism from $G$ to $G$, and $\phi$ is surjective, then $\phi(G) = G$. Therefore, $\phi(G)$ is cyclic and generated by $\phi(g)$. Hence, $\phi(g)$ is a generator of $G$.

I think that covers it. Let me make sure there's no gap here. The key points are that automorphisms are surjective homomorphisms, and the image of a generator under a surjective homomorphism must generate the image group, which in this case is $G$ itself. So yeah, $\phi(g)$ generates $G$. That should do it.

Proof:

Let $G$ be a cyclic group generated by an element $g$, and let $\phi : G \to G$ be an automorphism. We aim to show that $\phi(g)$ is also a generator of $G$.

1. Properties of Automorphisms: Since $\phi$ is an automorphism, it is a bijective homomorphism. Bijectivity ensures $\phi$ is surjective, meaning $\phi(G) = G$. Additionally, homomorphisms preserve group operations, so for any integer $k$,

$$\phi(g^k) = \phi(g)^k.$$

2. Image of $G$ Under $\phi$: As $\phi$ is surjective, every element $h \in G$ can be expressed as $h = \phi(a)$ for some $a \in G$. Since $G$ is cyclic and generated by $g$, there exists an integer $m$ such that $a = g^m$. Thus,

$$h = \phi(g^m) = \phi(g)^m.$$

3. Generating $G$ via $\phi(g)$: The equality $h = \phi(g)^m$ demonstrates that every element $h \in G$ is a power of $\phi(g)$. Consequently, $\phi(g)$ generates $G$, making $\phi(g)$ a generator of $G$.

4. Conclusion: Therefore, the automorphism $\phi$ maps the generator $g$ to another generator $\phi(g)$, preserving the cyclic structure of $G$.

$$\square$$

**MDTs ($L$)**

- Homomorphic Image of Cyclic Group is Cyclic Group
- Definition:Group Automorphism
- Automorphism Maps Generator to Generator
- Definition:Cyclic Group/Generator
- Definition:Group homomorphism

---

## A.11 THE SYSTEM PROMPT FOR BASELINE LLMS

---

**The system prompt for Baseline LLMs**

You are an AI assistant specialized in providing detailed mathematical solutions and summarizing theorems or concepts involved in the process. Your task is to present a complete set of steps for solving a problem or a summary of descriptive text, explicitly identifying and labeling any theorems or concepts used. Follow these guidelines for your output: 1)If a theorem is used, label it with ## before and after the theorem name. If a concept is used, prefix the concept name with "Definition:" and enclose it with ##.
Here are some examples of how to format your output:
e.g.:
input: {example}
output: {output_example}
DO NOT include any explanations or additional text in your output, just output the theorems or definitions separate by "##". Here are new mathematical solutions you need to extract and summarize:

---

## A.12 THE LLM-AS-JUDGE PROMPT FOR MDTs MATCH (ID EVALUATION)

---

**The LLM-as-Judge prompt for MDTs match (ID evaluation)**

You will receive two strings, A and B, each containing a list of theorem names or mathematical concepts separated by ##. Your task is to output a series of similar pairs, where:

- The left item of each pair comes from A.
- The right item of each pair comes from B.

Similarity Criteria:

- Match terms with identical or near-identical names (e.g., "Semilattice", "Semilattice")
- Pair items where one is a variant/subset of the other (e.g., "Closure", "Closure (Abstract Algebra)").
- Ignore minor formatting differences (e.g., "Idempotence", "Idempotence/Operation").
- Prioritize conceptual alignment over exact wording (e.g., "Associativity", "Associative").

Output Format:
Return a list of pairs in the format [[$A_{item}$, $B_{item}$]], one per line. If no match is found for an item, omit it.
Example:
Input:
A:

- Definition:Upper Semilattice
- Definition:Semilattice
- Definition:Closure
- Definition:Associativity
- Definition:Commutativity
- Definition:IdempotenceOperation
- Definition:Supremum of Set

B:

- Definition:Associative
- Definition:Algebraic Structure
- Definition:IdempotenceOperation
- Definition:Closure (Abstract Algebra)Algebraic Structure
- Definition:Upper Semilattice on Classical Set
- Definition:CommutativeOperation
- Upper Semilattice on Classical Set is Semilattice

Output:

- [[Definition:Upper Semilattice, Definition:Upper Semilattice on Classical Set]]
- [[Definition:Semilattice, Definition:Semilattice ]]
- [[Definition:Closure, Definition:Closure (Abstract Algebra)Algebraic Structure]]
- [[Definition:Associativity, Definition:Associative ]]
- [[Definition:Commutativity, Definition:CommutativeOperation]]
- [[Definition:Idempotence, Definition:IdempotenceOperation]]

Do not output any additional text or explanations, just the pairs. Here are the two termstheorems to compare:

---

For example, if the MDTs in labels are "Definition:A, B, Definition:C", and the MDTs in output are "Definition:A, B". The LLM judge output the matched MDTs: "Definition:A, B", then the precision is $2/2 = 1$, the recall is $2/3$, and the F1 score is $2 \cdot (1 \cdot 2/3)/(1 + 2/3) = 4/5$.

## A.13  THE LLM-AS-JUDGE PROMPT FOR ACCURACY MATCH (OOD EVALUATION)

---

**The LLM-as-Judge prompt for accuracy match (OOD evaluation)**

Act as a mathematics expert. You will receive two strings, A and B. B string contains a list of theorem names or mathematical concepts separated by `##`.

Given two mathematical sets:

- **A:** A mathematical domain/topic (e.g., "Linear Algebra," "Topology," "Calculus").

- **B:** A list of mathematical theorems, concepts, or terms.

Perform the following:

1. Check if each item in B belongs to the domain A (based on standard mathematical classifications).

2. Return all strings from B that are definitively part of A, preserving their original formatting (e.g., capitalization, symbols).

**Additional Notes:**

- If no items in B match A, output `None`.

**Output Format:** Select the items from B that are part of A, separated by `##`.

**Example:**

- **Input:**
  A:`<|Mathematics -> Precalculus -> Limits|>` and
  B:`<|L'Hôpital's Rule##Definition:Limit of Real Function|>`
  **Output:** `L'Hôpital's Rule##Definition:Limit of Real Function`

- **Input:**
  A:`<|Mathematics -> Algebra -> Intermediate Algebra ->`
  `Quadratic Functions|>` and
  B:`<|Solution to Quadratic Equation##Definition:Real`
  `Number##Definition:Domain (Set)|>`
  **Output:**
  `Solution to Quadratic Equation##Definition:Real Number`
  `##Definition:Domain (Set)`

- **Input:**
  A:`<|Mathematics -> Algebra -> Linear Algebra|>` and
  B:`<|Euler's Formula##Determinant##Fourier`
  `Transform##Eigenvector##Riemann Hypothesis|>`
  **Output:** `Determinant##Eigenvector`

- **Input:**
  A:`<|Mathematics -> Geometry -> Solid Geometry -> Surface`
  `Area|>` and
  B:`<|Definition:Angle##Definition:Area##Definition:Circle/Center|>`
  **Output:** `Definition:Angle##Definition:Area`
  `##Definition:Circle/Center`

- **Input:**
  A:`<|Mathematics -> Geometry -> Solid Geometry -> Surface`
  `Area|>` and
  B:`<|Intermediate Value Theorem|>`
  **Output:** `None`

Here are the two terms/theorems to compare:

---

## A.14 THE SYSTEM PROMPT REASONMAP MODEL

---

**The system prompt ReasonMap models**

---

You are an AI assistant specialized in providing detailed mathematical solutions and summarizing theorems or concepts involved in the process. Your task is to present a complete set of steps for solving a problem or a summary of descriptive text, explicitly identifying and labeling any theorems or concepts used. Follow these guidelines for your output: 1)If a theorem is used, label it with ## before and after the theorem name. If a concept is used, prefix the concept name with "Definition:" and enclose it with ##.

---

## A.15 ERROR CASES AFTER STAGE 2

**Case 1:** Incomplete prediction

- Model output: Laplace Transform of Generating Function
- Label: Laplace Transform of Generating Function of Sequence

**Case 2:** Overly general prediction

- Model output: Existence-Uniqueness Theorem for First-Order Differential Equation
- Label: Existence-Uniqueness Theorem for Homogeneous First-Order Differential Equation

**Case 3:** Relevant but incorrect prediction

- Model output: Natural Number Multiplication is Cancellable
- Label: Non-Zero Integers are Cancellable for Multiplication

# B  EXTENDED EXPERIMENTAL RESULTS

## B.1  PERFORMANCE ON NEXT-GENERATION MODELS

To validate the adaptability of our framework to the rapid iteration of open-weight LLMs, we applied ReasonMap to a newer base model: **Qwen3-8B**. Unlike the instruction-tuned variants used in our main experiments, Qwen3-8B is a base model without inherent instruction-following capabilities. Despite this, after training with our ReasonMap pipeline, it demonstrates exceptional performance.

Table 5 presents the results. The **ReasonMap-8B**, using Qwen3-8B as base model, achieves better results across all metrics, particularly in F1-score (0.860 on ID) and OOD generalization. This confirms that our training framework is not only robust but also effectively unlocks the potential of latest-generation foundation models for fine-grained knowledge extraction.

Table 5: Performance comparison of ReasonMap across different backbone architectures. The integration with the newer Qwen3-8B base model yields significant improvements in both F1-score and Precision, validating the framework's scalability to next-generation models.

| Models | In distribution (ID) | | | | | | Out-of-distribution (OOD) | |
|---|---|---|---|---|---|---|---|---|
| | $\mathcal{D}^{test}_{acq}$ | | | $\mathcal{D}^{test}_{long}$ | | | $\mathcal{D}^{test}_{DeepMath}$ | $\mathcal{D}^{test}_{Omni}$ |
| | P | R | F1 | P | R | F1 | ACC | ACC |
| Qwen2.5-7B-Instruct | | | | | | | | |
| Base-Model | 0.610 | 0.611 | 0.610 | 0.443 | 0.580 | 0.502 | 0.412 | 0.401 |
| ReasonMap-7B$_{dpo}$ | 0.786 | 0.766 | 0.775 | 0.732 | 0.737 | 0.734 | 0.455 | 0.413 |
| Llama3.1-8B-Instruct | | | | | | | | |
| Base-Model | 0.577 | 0.649 | 0.611 | 0.359 | 0.702 | 0.475 | 0.321 | 0.365 |
| ReasonMap-8B$_{dpo}$ | 0.758 | 0.742 | 0.749 | 0.789 | 0.715 | 0.750 | 0.432 | 0.389 |
| Qwen3-8B | | | | | | | | |
| ReasonMap-8B$_{dpo}$ | 0.864 | 0.858 | 0.860 | 0.767 | 0.759 | 0.763 | 0.469 | 0.400 |

## B.2  DOWNSTREAM UTILITY ON STRONG REASONING MODELS

To further validate the utility of extracted MDTs (Section 4.5), we conducted additional experiments using the challenging **AIMO Validation Set** (Math Olympiad level). We evaluated whether providing MDTs as hints improves the performance of both a strong reasoning model (**DeepSeek-V3**) and a smaller open weights model (Qwen2.5-7B-Instruct).

As shown in Table 6, explicitly injecting MDTs yields consistent performance gains across both model classes. Notably, DeepSeek-V3 achieves a **5.56%** improvement, suggesting that even SOTA reasoning models benefit from precise knowledge grounding.

| Model | Base Accuracy | With MDTs | Improvement ($\Delta$) |
|---|---|---|---|
| Qwen2.5-7B | 10.00% | 14.44% | +4.44% |
| DeepSeek-V3 | 51.11% | 56.67% | **+5.56%** |

Table 6: Downstream performance on the AIMO validation set. Providing extracted MDTs as contextual hints improves accuracy for both weak and strong reasoning models.

# C  DOWNSTREAM APPLICATION VIA SELF-GENERATED MDTS

In Section 4.5, we utilized ground-truth MDTs (Oracle) to establish a theoretical upper bound for how structural knowledge can aid reasoning. To further demonstrate the practical utility of Reason-Map in a realistic, inference-only setting where ground truth is unavailable, we introduced a new experiment.

This experiment operates entirely on the model's own generations. Specifically, for a given problem, we first sample $k = 4$ candidate reasoning paths using the base model (Qwen2.5-7B-Instruct). Our fine-tuned ReasonMap-7B model is then applied to extract MDTs from each of these generated paths. These extracted knowledge points are aggregated and further filtered via the MinHash algorithm to minimize redundancy. The resulting refined set ($H_{self}$) is subsequently provided back to the base model as contextual hints for a final inference pass. This pipeline effectively acts as a self-correction mechanism, distilling structured knowledge from the model's initial exploration to reinforce its final reasoning.

We evaluated this strategy on the Math 500 and DeepMath-1K testsets. As presented in Table 7, the results confirm that our method provides tangible benefits even without access to ground-truth solutions. The "Self-Generated" setting consistently outperforms the base model baseline, achieving an accuracy of **75.1%** on Math 500 (vs. 74.0%) and **51.3%** on DeepMath-1K (vs. 50.6%). While the Oracle setting (75.9% and 51.8%) remains the theoretical ceiling.

Table 7: Performance comparison on Qwen2.5-7B-Instruct using different sources of MDT hints. The "Self-Generated" setting, which relies solely on MDTs extracted from the model's own rollouts ($k = 4$), consistently outperforms the base model baseline, demonstrating practical utility in realistic inference scenarios.

| Method Setting | Math 500 (Acc) | DeepMath-1K (Acc) |
|---|---|---|
| **Qwen2.5-7B-Instruct (Base Model)** | 0.740 | 0.506 |
| **+ Self-Generated MDTs** | **0.751** | **0.513** |
| *+ Oracle MDTs* | *0.759* | *0.518* |

# D  HUMAN EVALUATION AND QUALITATIVE ANALYSIS

We conducted a series of human evaluations involving three PhDs in Mathematics to validate our automated metrics and analyze model behavior.

## D.1  VALIDATION OF LLM-AS-A-JUDGE

Table 8: Full breakdown of agreement scores between the GPT-4o-mini judge and three independent human experts. The automated judge achieves consistently high Precision and Recall across all individual annotators, demonstrating robustness.

| Evaluation Task | Human Expert | Precision | Recall |
|---|---|---|---|
| **Domain Relevance** | Annotator A | 0.9083 | 0.9033 |
| | Annotator B | 0.9100 | 0.9070 |
| | Annotator C | 0.9050 | 0.9080 |
| | *Average* | *0.9078* | *0.9061* |
| **Fine-grained MDTs** | Annotator A | 0.9240 | 0.9330 |
| | Annotator B | 0.9190 | 0.9280 |
| | Annotator C | 0.9300 | 0.9260 |
| | *Average* | *0.9243* | *0.9290* |

To ensure the reliability of GPT-4o-mini as our automated judge, we constructed a validation set by randomly sampling 100 instance pairs from our test outputs, including 50 fine-grained MDT pairs from the In-Distribution (ID) set and 50 domain label pairs from the Out-of-Distribution (OOD) set. The human evaluators independently assessed these pairs for semantic equivalence, and their matching results are used to calculate the precision and recall with the results judged by GPT-4o-mini. The automated judge exhibited strong alignment with human experts, achieving high Average Precision and Recall scores ($> 0.90$) across both fine-grained MDT matching and OOD domain

categorization. These consistent high-agreement results confirm the effectiveness of GPT-4o-mini in capturing semantic correctness, validating its use as a reliable proxy for AAMDT task evaluation.

## D.2 Error Analysis and Taxonomy

We performed a fine-grained error analysis on 100 failure cases (where overlap $< 50\%$) from both DeepSeek-V3 and ReasonMap-7B. Errors were classified into three categories:

- **Code A (Granularity Mismatch):** The predicted concept is correct, but the level of detail/specificity is inconsistent with the label. The prediction is a generalization of the label, or the label is a generalization of the prediction.
- **Code B (Implicit Omission):** The model fails to extract one or more critical and non-trivial MDTs that are present in the Ground Truth Label. (Focus on key theorems or domain-specific definitions that are entirely missing).
- **Code C (Inference Drift / Hallucination):** The model predicts MDTs that are clearly irrelevant or semantically distant from the actual core concepts in the Ground Truth Label. (The prediction includes concepts that share very little semantic overlap with any item in the label).

Table 9 shows the distribution. The most critical success of ReasonMap lies in the suppression of Inference Drift (Code C) errors. These errors are considered the most destructive as they introduce misleading or semantically distant concepts into the annotation. DeepSeek-V3's primary failure mode is Code C, accounting for **58%** of its total errors. ReasonMap successfully reduces this high-risk failure mode by 10 percentage points (from 58% to **48%**). This demonstrates that our training strategy (SFT+DPO) effectively suppresses the generation of noisy and unreliable concepts, preventing the model from "hallucinating" connections that do not logically exist.

The error distribution in ReasonMap shifts towards categories that are more recoverable and less harmful: 1) Granularity Mismatch (Code A): These errors are semantically controllable (i.e., "correct concept, wrong specificity"). ReasonMap is slightly more prone to this recoverable error type (12% vs. 10%), indicating it stays "on topic" even when the precision is imperfect. 2) Implicit Omission (Code B): The increase in omissions (40% vs. 32%) suggests a more conservative strategy. We acknowledge this increase in Code B as a side effect of our alignment algorithm. In the process of suppressing excessive model output to reduce hallucinations, the model effectively raises its internal confidence threshold, which inevitably increases the probability of missing some valid but less obvious key information.

This targeted error analysis confirms that ReasonMap does not merely improve overall F1 scores; it fundamentally improves the quality and safety of model outputs during failure cases. By significantly reducing the most unreliable error pattern (Code C)—even at the cost of increased omissions.

Table 9: Distribution of error types. ReasonMap significantly reduces severe "Inference Drift" (Code C) errors compared to the baseline.

| Model | Code A (Granularity) | Code B (Omission) | Code C (Drift) |
|---|---|---|---|
| DeepSeek-V3 | 10% | 32% | 58% |
| ReasonMap | 12% | 40% | 48% |

## D.3 Evaluation of OOD Triviality

To address concerns that the automated "Domain Relevance" metric (judged by GPT-4o-mini) might artificially inflate performance by including many trivial tags (e.g., "Definition: Set"), we conducted a targeted audit to determine the triviality rate within the accepted predictions.

Specifically, we sampled 300 extracted MDTs that were already classified by GPT-4o-mini as belonging to the given domain. Human experts then annotated these validated MDTs to identify those that were "trivial" or "uninformative" given the problem context. The evaluation reveals that only **18.3%** of the MDTs were classified as trivial. This low ratio confirms that the high domain relevance scores reported in our main results largely reflect the retrieval of non-trivial, substantial mathematical knowledge, rather than generic or redundant concepts.

# E    DATA CONSTRUCTION DETAILS

In Stage 1, we augment the dataset by prompting an LLM to generate long-form CoTs ($A_l$) based on concise human proofs ($A$). A potential concern is whether the model simply copies or paraphrases the short proof ("Over-Imitation") rather than generating a genuinely detailed reasoning path.

The divergence between Precision and Recall provides critical insights into the generation quality: 1) The low Precision scores (e.g., ROUGE-1 $\approx 0.11$) are expected and desirable. Since the generated CoTs are approximately $30\times$ longer than the original proofs (expanding from $<0.1$k to $\approx 3$k tokens), the original tokens constitute only a small fraction of the generated text. This confirms that the model is *not* simply copying the input, but is performing substantial. 2) Conversely, the relatively high Recall scores (e.g., ROUGE-1 $\approx 0.74$) indicate that the vast majority of key terms and steps from the original concise proof ($A$) are successfully retained within the generated long CoT ($A_l$). The results show the augmentation faithfully diversifies the reasoning data and maintains the correct answer, instead of introducing noise and bias.

Table 10: ROUGE scores between original concise proofs and generated long CoTs. The low overlap indicates significant structural and content expansion.

| Metric | Precision | Recall | F1-Score |
|--------|-----------|--------|----------|
| ROUGE-1 | 0.1089 | 0.7398 | 0.1771 |
| ROUGE-2 | 0.0578 | 0.4135 | 0.0948 |
| ROUGE-L | 0.0776 | 0.5720 | 0.1280 |

# F    COST-EFFECTIVENESS ANALYSIS

A key motivation for developing ReasonMap is to provide a scalable, cost-effective solution for fine-grained mathematical annotation. While proprietary models like the recently released **GPT-5-mini** offer competitive pricing, their costs scale linearly with dataset size, making them prohibitively expensive for annotating the massive corpora required for pre-training (e.g., millions of samples).

**Parameters and Assumptions.**

- **Workload:** We estimate costs for annotating 1 Million (1M) samples.

- **Token Counts:** We assume an average input length of 2,000 tokens and an output length of 500 tokens per instance, totaling 2.5 billion tokens.

- **GPT-5-mini Pricing**[6]**:** $0.25 per 1M input tokens and $2.00 per 1M output tokens (August 2025 pricing).

- **ReasonMap Pricing:** Our cost model uses standard cloud pricing for NVIDIA A100 SXM GPUs ($1.79/hour/GPU[7]).

    - *Training:* We assume a fine-tuning run using 8 GPUs for 8 hours.
    - *Inference:* We assume deployment on a single GPU. With vLLM acceleration, the system achieves a throughput of $\sim$3,362 tokens/sec [8].

Based on these data, the inference cost is derived as follows:

$$\text{Total Time (seconds)} = \frac{2,500,000,000 \text{ tokens}}{3362.71 \text{ tokens/s}} \approx 743,448 \text{ s}$$

$$\text{Total Time (hours)} = \frac{743,448}{3,600} \approx 206.51 \text{ hours}$$

$$\text{Inference Cost} = 206.51 \text{ hours} \times \$1.79/\text{hr} \approx \mathbf{\$369.65}$$

---

[6]https://openai.com/api/pricing/
[7]https://lambda.ai/instances
[8]https://www.databasemart.com/blog/vllm-gpu-benchmark-a100-80gb

Table 11 presents the economic analysis. While the proprietary API incurs linear costs reaching $1,500, the self-hosted approach—even accounting for an 8-GPU training cluster—costs significantly less.

Table 11: Cost comparison for annotating 1 million samples. Even with the upfront cost of training on an 8-GPU cluster, the open-source ReasonMap model reduces the total expense by **approx. 68%** ($484 vs. $1,500) compared to the proprietary frontier API. Inference costs are calculated based on a measured throughput of 3,362 tokens/s.

| Method | Cost Breakdown (1M Samples) | | Total Cost |
|---|---|---|---|
| | Operational Components | Cost Calculation | |
| **GPT-5-mini (API)** | Input Processing | 2,000M tok × $0.25 | $500.00 |
| | Output Generation | 500M tok × $2.00 | $1,000.00 |
| | | *API Total:* | **$1,500.00** |
| **ReasonMap-7B (Ours)** | Training (8x A100) | 8 hours × 8 × $1.79 | $114.56 |
| | Inference (1x A100) | ∼207 hours × $1.79 | $369.66 |
| | | *Self-Hosted Total:* | **$484.22** |

ReasonMap demonstrates a clear economic advantage for large-scale data synthesis. The combined cost of training and inference (∼$484) is substantially lower than the fees for a comparable volume of API calls. This allows researchers to trade variable API costs for fixed compute time, making massive dataset generation feasible within academic budgets.

