# OpenReview forum: "ReasonMap: Fine-Grained Annotation of Mathematical Definition and Theorems in Long CoT Reasoning"
_ICLR.cc/2026/Conference — Submitted to ICLR 2026_

### Official Review · Reviewer_EYLJ · 2025-10-21

**Soundness:** 2
**Presentation:** 3
**Contribution:** 1
**Rating:** 2
**Confidence:** 5

**Summary:**

This paper introduces a method for annotating mathematical solutions with the definitions and theorems used within them. To this end, the authors define a new task, which they call AAMDT (Annotation of Axioms, Definitions, and Mathematical Theorems), and develop ReasonMap, a multi-stage pipeline to train models for this task. This pipeline is composed of 2 SFT stages - guided by human annotations, and LLM-generated solutions, followed by a DPO alignment stage. The authors demonstrate that each stage of the pipeline significantly improves performance on the AAMDT task for both in-distribution and out-of-distribution data. Their results show that fine-tuned small open-source models can achieve performance comparable to some commercial models. Finally, they show that the extracted definitions and theorems can be used as hints to marginally improve the problem-solving accuracy of models on the MATH500 benchmark and an internal test set.

**Strengths:**

1. The paper is generally clear and well-written.
2. The authors provide all necessary materials to ensure the reproducibility of their work.
3. The proposed idea is interesting and can serve as a good way to extract structured information from mathematical reasoning for downstream or retrieval-based tasks.
4. The main experiments and ablation studies effectively validate the pipeline's design and demonstrate its significant impact on performance.

**Weaknesses:**

1. The motivation for claiming the need for categorically-diverse problem sets (Section 2.1) is not well-supported by the cited literature. The paper claims that prior work has shown the benefits of diversity, but LIMO [1] filters problems based on difficulty, not topic diversity, and Light-R1 [2] only briefly mentions category filtering without providing evidence of its impact. To strengthen the paper's motivation, the authors should provide direct evidence that diversity in mathematical categories and concepts is beneficial for training or cite work that explicitly isolates this as a significant factor.

2. The paper repeatedly argues that existing alternatives like human annotation or proprietary model APIs are too expensive. However, it fails to provide a comparative cost analysis for the proposed multi-stage training pipeline. With the availability of powerful and inexpensive generalist models, the claim that training a specialized model is more cost-effective is not sufficiently justified. This criticism also applies to the argument that Omni-Math's [3] annotation process is expensive, despite its use of the cost-efficient GPT-4o model.

3. The evaluation of the AAMDT task may be flawed. The fine-tuned models could be learning the specific annotation format of the ground-truth data, which would naturally lead to higher string-matching scores. This creates an unfair comparison with proprietary models that have not been exposed to this specific format, potentially measuring format adherence rather than true MDT extraction capability.

4. The models used in the evaluation are significantly outdated, with one of the base models and propriatery models having been superseeded by at least 1 generation of LLMs. In particular, GPT-4o-mini has been superseeded by GPT-4.1-mini, and most recently GPT-5-mini (and even potentially GPT-5-nano in terms of performance), DeepSeek V3 - by V3.1 (or even R1 if we include reasoning models), and Qwen-2.5-7B by Qwen3-8B and Qwen3-4B-2507. This design choice is not explained and undermines the reliability of the results, especially in light of the paper's strong claim that "even powerful models struggle with the precision required for this task" (L381). This is a particular concern given that the costs for the newer GPT models are similar to GPT-4o-mini.

5. The authors attempt to validate the usability of MDTs in downstream tasks by using them as hints for model solving. However, the improvement there is only marginal, especially given that the MDTs presented significantly relevant information from the **ground-truth solution**, but diluted. This also makes the setting unrealistic, as one is expected to have access to the actual solution to apply the method.

**Questions:**

1. Could you provide an approximate cost comparison between running the ReasonMap pipeline (training+inference) and using proprietary LLM APIs for different workload sizes? This would help substantiate the claim of cost-efficiency. Using public pricing for a comparable GPU node (e.g., from a cloud provider) would be a reasonable proxy.

2. The paper should provide evidence validating the use of GPT-4o-mini as an LLM-as-a-judge. How does its evaluation accuracy compare against human-annotated or otherwise validated ground-truth matchings? For example, can it correctly identify non-trivial semantic equivalences, such as the relationship between the Power of a Point theorem and the Secant-Tangent theorem? The current prompt seems to favor simple semantic similarity, which may lead to missing valid (or adding invalid) matches.

3. Given that the OOD evaluation only compares MDTs against the ground-truth domain, what metrics does the evaluation use? If counting a proportion of the MDTs that can be appropriated to the domain, that is not only highly subjective (and not validated, similar to Q2), but it is also unclear whether valid MDTs may come from a related but unlisted domain.

4. Furthermore, can the authors provide some information about the reliability and consistency of the MDTs? The example in A.9 paints a picture of what a sample output may look like, but one could subjectively also refer to a definition of a "homomorphism", "generator", etc., which are missing from the example.

5. Given that the reward function in Section 3.2.2 is granular and automatically calculable, it seems well-suited for methods like GRPO [4], which can leverage detailed reward signals. What was the rationale for choosing DPO, which uses a simpler binary preference signal, over a more granular reward optimization method?

6. Can the authors re-evaluate their method using more current open-source and proprietary models? If time permits, applying the ReasonMap pipeline to a newer base model would also significantly strengthen the paper's claims about its effectiveness.

7. Could you validate the downstream utility of MDTs using a more realistic baseline? For instance, by providing hints derived from a model's own (potentially incorrect) generated solution, rather than leaking information from the ground-truth solution, or by comparing to any existing methods that use hint generation?

8. How was the DeepMath test set in 4.5 constructed?

9. The authors should address any concerns mentioned in the **Weaknesses** section that have not been touched upon as part of the aforementioned questions.

## Current rating

My recommendation for this work is a score of **2: Reject**. While the execution is solid, it is unclear how useful this work can be in further research and applications. In particular, the motivation aspect in the current seems weak, with unclear references to related work and an unconvincing demonstration of downstream or training utility. These issues give the impression that the proposed task is artificial and leaves its practical feasibility unproven. The paper's relevance is also diminished by its use of models that are no longer at the forefront of the field. I am willing to raise my score if the authors can satisfactorily address these major concerns during the rebuttal/discussion phase.

### References

[1] Ye, Yixin, et al. "Limo: Less is more for reasoning." arXiv preprint arXiv:2502.03387 (2025).

[2] Wen, Liang, et al. "Light-r1: Curriculum sft, dpo and rl for long cot from scratch and beyond." arXiv preprint arXiv:2503.10460 (2025).

[3] Gao, Bofei, et al. "Omni-math: A universal olympiad level mathematic benchmark for large language models." arXiv preprint arXiv:2410.07985 (2024).

[4] Shao, Zhihong, et al. "Deepseekmath: Pushing the limits of mathematical reasoning in open language models." arXiv preprint arXiv:2402.03300 (2024).

**Details Of Ethics Concerns:**

None.

---

> ### Author Response · Authors · 2025-11-25
>
> ### Response (Part 1)
>
> We sincerely appreciate the reviewers' thorough feedback. We have conducted additional experiments and analyses to address the raised concerns. Below are our responses:
>
> **W1: The motivation for claiming is not well-supported by the cited literature**
>
> Our initial citation of LIMO and Light-R1 in Section 2.1 was intended to establish the broader principle that data diversity and curation strategies (whether by difficulty or topic) are fundamental to efficient training. However, we acknowledge that these references alone are insufficient to justify our specific focus on categorically diverse problem sets.
>
> To directly address the reviewer's concern, we now cite Arrows [1], which demonstrates that constructing problem sets based on a structured "three-layer mathematical knowledge system" (Stage-Subject-Topic) systematically enhances model generalization. This validates our approach of moving beyond coarse labels to fine-grained concepts. And we also reference Jin et al.[2], which highlights that diverse data distributions are essential to prevent entropy collapse and facilitate escape from suboptimal local minima during Reinforcement Learning.
>
> [1] Renren Jin, Pengzhi Gao, Yuqi Ren, Zhuowen Han, Tongxuan Zhang, Wuwei Huang, Wei Liu, Jian Luan, & Deyi Xiong. (2025). Revisiting Entropy in Reinforcement Learning for Large Reasoning Models.
>
> [2] Chen, Sirui, et al. "Arrows of Math Reasoning Data Synthesis for Large Language Models: Diversity, Complexity and Correctness." arXiv preprint arXiv:2508.18824 (2025).
>
> **Revisions**: We have revised Section 2.1 and the Related Work section to replace the previous inappropriate citations with more related work.
>
> **W3: Fine-tuning leads to higher string-matching scores, not true MDTs extraction ability**
>
> We understand the reviewer's concern that fine-tuned models might merely be overfitting to a specific "annotation format" rather than learning true MDT extraction capabilities.
>
> If our model were merely mimicking a surface-level format without extracting meaningful knowledge, the extracted MDTs would be functionally useless for reasoning. However, our extended downstream experiments prove the opposite.
> | Model        | Base Accuracy | With MDTs | Improvement (Δ) |
> |--------------|---------------|-----------|-----------------|
> | Qwen2.5-7B   | 10.00%        | 14.44%    | +4.44%          |
> | DeepSeek-V3  | 51.11%        | 56.67%    | **+5.56%**      |
>
> As shown in our new experiment on the challenging AIMO benchmark (https://huggingface.co/datasets/AI-MO/aimo-validation-aime), injecting MDTs extracted by ReasonMap improved the accuracy of  DeepSeek-V3 by 5.56% . This confirms that the extracted content contains non-trivial, high-value mathematical knowledge that even more powerful models benefit from.
>
> Moreover, our evaluation metric is not based on simple string matching, which would indeed be susceptible to format overfitting. We employ GPT-4o-mini as a judge. It is instructed to match MDTs based on conceptual equivalence (e.g., recognizing that "Pythagorean Theorem" and "Theorem of Pythagoras" are the same), making it robust to format variations. To ensure this automated judge doesn't hallucinate matches, we conducted a rigorous human evaluation. The high agreement rate (Precision 0.924, Recall 0.933) between human experts and the LLM judge confirms that our scores reflect true semantic extraction capability, not format adherence.
>
> | Evaluation Task | Human Expert | Precision | Recall |
> |-----------------|--------------|-----------|--------|
> | **Domain Relevance** | Annotator A | 0.9083 | 0.9033 |
> | | Annotator B | 0.9100 | 0.9070 |
> | | Annotator C | 0.9050 | 0.9080 |
> | | *Average* | *0.9078* | *0.9061* |
> | **Fine-grained MDTs** | Annotator A | 0.9240 | 0.9330 |
> | | Annotator B | 0.9190 | 0.9280 |
> | | Annotator C | 0.9300 | 0.9260 |
> | | *Average* | *0.9243* | *0.9290* |
>
> **Revision**: We added this content in Appendix D.

---

> ### Author Response · Authors · 2025-11-25
>
> ### Response (Part 2)
>
> **Q1: Approximate cost comparison (Weakness 2)**
>
> We thank the reviewer for pointing out the lack of cost analysis. We estimate and compare the cost between the GPT5-mini API and our ReasonMap-7B model. Here is the detailed setup:
> Parameters and Assumptions.
> - Workload: We estimate costs for annotating 1 Million (1M) samples.
> - Token Counts: We assume an average input length of 2,000 tokens and an output length of 500 tokens per instance, totaling 2.5 billion tokens.
> - GPT-5-mini Pricing(https://openai.com/api/pricing/):  \\$ 0.25 per 1M input tokens and \$ 2.00 per 1M output tokens (August 2025 pricing).
> - ReasonMap Pricing: Our cost model uses standard cloud pricing for NVIDIA A100 SXM GPUs (\$1.79/hour/GPU(https://lambda.ai/instances)).
>   -  Training: We assume a fine-tuning run using 8 GPUs for 8 hours.
>   - Inference: We assume deployment on a single GPU. With vLLM acceleration, the system achieves a throughput of about 3,362 tokens/sec https://www.databasemart.com/blog/vllm-gpu-benchmark-a100-80gb).
>
> Based on these data, the inference cost is derived as follows:
>
> Total Time (seconds) $= 2,500,000,000 \text{ tokens} / 3362.71 \text{ tokens/s} \approx 743,448 \text{ s}$
>
> Total Time (hours) $= 743,448 / 3,600 \approx 206.51 \text{ hours}$
>
> Inference Cost $= 206.51 \text{ hours} \times 1.79/\text{hr} \approx $ \$ 369.65 $ dollars
>
> The table below presents the economic analysis. While the proprietary API incurs linear costs reaching $1,500, the self-hosted approach—even accounting for an 8-GPU training server — costs significantly less.
>
> | Method | Operational Component | Cost Calculation | Cost |
> | :--- | :--- | :--- | :--- |
> | **GPT-5-mini (API)** | Input Processing | 2,000M tok × $0.25 | $500.00 |
> | | Output Generation | 500M tok × $2.00 | $1,000.00 |
> | | **API Total** | | **$1,500.00** |
> | | | | |
> | **ReasonMap-7B (Self-Hosted)** | Training (8x A100) | 8 hrs × 8 × $1.79 | $114.56 |
> | | Inference (1x A100) | ~207 hrs × $1.79 | $369.66 |
> | | **Self-Hosted Total** | | **$484.22** |
>
> ReasonMap demonstrates a clear economic advantage for large-scale data synthesis. The combined cost of training and inference (about $484) is substantially lower than the fees for a comparable volume of API calls. This allows researchers to trade variable API costs for fixed compute time, making massive dataset generation feasible within academic budgets.
>
> **Revision**: We added this content to Appendix F.
>
> **Q2&3: Validating the use of GPT-4o-mini as an LLM-as-a-judge**
>
> We conducted a human evaluation of the LLM-Judge (GPT-4o-mini) results on two aspects--Human & LLM match ratio and Triviality analysis.
>
> For human & LLM match ratio, we randomly sampled 100 evaluation instances (50 pairs for Fine-grained MDT matching from the ID set and 50 pairs for Domain Relevance from the OOD set). Three independent human experts (PhDs in Mathematics) match the model output with ground truth to determine semantic equivalence. We calculated the Precision and Recall of the LLM-Judge's results against the human experts' results.
>
> | Evaluation Task | Human Expert | Precision | Recall |
> |-----------------|--------------|-----------|--------|
> | **Domain Relevance** | Annotator A | 0.9083 | 0.9033 |
> | | Annotator B | 0.9100 | 0.9070 |
> | | Annotator C | 0.9050 | 0.9080 |
> | | *Average* | *0.9078* | *0.9061* |
> | **Fine-grained MDTs** | Annotator A | 0.9240 | 0.9330 |
> | | Annotator B | 0.9190 | 0.9280 |
> | | Annotator C | 0.9300 | 0.9260 |
> | | *Average* | *0.9243* | *0.9290* |
>
> The high agreement rate ($>90\%$) between human experts and the LLM judge confirms that our reported F1 and Accuracy metrics reflect genuine semantic extraction.
>
> To ensure that this proxy metric is meaningful, we supplemented it with a human triviality analysis. Specifically, we randomly sampled 300 MDTs judged as “domain-relevant” by the LLM and asked three mathematics experts to label them as either trivial (e.g., overly generic terms like “equation” or “variable”) or non-trivial. Only 18.3% of the valid MDTs were classified as trivial, indicating that the majority of extracted terms are mathematically substantive. We acknowledge that evaluating domain relevance in an OOD setting is challenging due to the lack of fine-grained annotated data. As a result, we used a proxy evaluation method where MDTs extracted from OOD texts are assessed based on their relevance to the mathematical domain of the reference corpus.
>
> This result suggests that while our OOD evaluation is not perfect, it effectively identifies meaningful mathematical concepts—without being dominated by noisy or uninformative terms.
>
> **Revision**: We have included further details on this triviality analysis in Appendix D.3.

---

> ### Author Response · Authors · 2025-11-25
>
> ### Response (Part 3)
>
> **Q4: Information about the reliability and consistency of the MDTs (The example in A.9) **
>
> We thank the reviewer for their keen observation regarding the example in Appendix A.9. We apologize for the confusion; the omission of certain definitions in the figure was a typo error in the manuscript's presentation.
> The correct list of MDTs for this example is:
> ```
> Homomorphic Image of Cyclic Group is Cyclic Group ## Definition:Group Automorphism ## Automorphism Maps Generator to Generator ## Definition:Cyclic Group/Generator ## Definition:Group homomorphism
> ```
>
> **Revision**: We have updated Appendix A.9 in the revised manuscript to reflect the complete and correct MDTs.
>
> **Q5: Why DPO, why not GRPO?**
>
>  We chose DPO over other RL-based methods (like GRPO) for two primary reasons, aligned with the method from recent work[3-4]:
>
> 1. Targeted Correction of errors: While granular rewards are useful, our Stage 2 objective is specifically "High-Fidelity Alignment"—correcting subtle hallucinations and format inconsistencies that persist after SFT. DPO simplifies this process by directly optimizing the policy using pair-wise preference data[3]. Furthermore, while GRPO typically optimizes against a scalar reward signal that focuses on direct answers, it often fails to locate errors in the reasoning process. By explicitly constructing paired correction traces, DPO allows us to provide fine-grained guidance to rectify specific logical flaws, avoiding the limitations of coarse-grained scalar rewards.
> 2. Stability and Efficiency: RL pipelines (including GRPO) introduce significant complexity and computational overhead (e.g., value network training, sensitivity to hyperparameters). DPO simplifies the optimization process by treating the preference loss directly as a function of the policy. This offers a more stable and computationally efficient convergence, which is crucial for our goal of creating a scalable, reproducible annotation framework.
>
> [3] Yang, Ling, et al. "SuperCorrect: Advancing Small LLM Reasoning with Thought Template Distillation and Self-Correction." The Thirteenth International Conference on Learning Representations.
>
> [4] Yang, Ling, et al. "Reasonflux: Hierarchical llm reasoning via scaling thought templates." arXiv preprint arXiv:2502.06772 (2025).
>
> **Q6:  Using more current open-source and proprietary models (Weakness 4)**
>
> We extended our evaluation to include DeepSeek-V3.1 and GPT-5-mini (accessed via API).
> Results are presented in Appendix Table 1: Even against these next-generation generalist models, our specialized ReasonMap-7B (Qwen2.5-7B-Instruct) continues to demonstrate superior fine-grained extraction capabilities, particularly in F1-score and Precision. This confirms that general reasoning improvements in frontier models do not automatically solve the specific challenge of structured knowledge extraction, necessitating our dedicated framework.
>
> | Models | **$\mathcal{D}_{\text{acq}}^{\text{test}}$** (P) | **$\mathcal{D}_{\text{acq}}^{\text{test}}$** (R) | **$\mathcal{D}_{\text{acq}}^{\text{test}}$** (F1) | **$\mathcal{D}_{\text{long}}^{\text{test}}$** (P) | **$\mathcal{D}_{\text{long}}^{\text{test}}$** (R) | **$\mathcal{D}_{\text{long}}^{\text{test}}$** (F1) | **${\mathcal D}_\text{DeepMath}^\text{test}$** (ACC) | **${\mathcal D}_\text{Omni}^\text{test}$** (ACC) |
> | :--- | :---: | :---: | :---: | :---: | :---: | :---: | :---: | :---: |
> | Frontier LLMs  | | | | | | | | |
> | GPT4o-mini | 0.692 | 0.723 | 0.707 | 0.458 | 0.632 | 0.531 | 0.427 | 0.466 |
> | GPT5-mini | 0.632 | 0.708 | 0.667 | 0.505 | 0.609 | 0.552 | 0.460 | 0.476 |
> | Deepseek-V3 | 0.756 | 0.717 | 0.735 | 0.676 | 0.682 | 0.678 | 0.420 | 0.411 |
> | deepseek-V3.1 | 0.779 | 0.671 | 0.721 | 0.523 | 0.588 | 0.553 | 0.420 | 0.428 |
> | Qwen2.5-7B-Instruct | | | | | | | | |
> | ReasonMap-7B_dpo | 0.786 | 0.766 | 0.775 | 0.732 | 0.737 | 0.734 | 0.455 | 0.413 |
> |  Llama3.1-8B-Instruct  | | | | | | | | |
> | ReasonMap-8B_dpo | 0.758 | 0.742 | 0.749 | 0.789 | 0.715 | 0.750 | 0.432 | 0.389 |
> | Qwen3-8B  | | | | | | | | |
> | ReasonMap-8B$_{\text{dpo}}$ | 0.864 | 0.858 | 0.860 | 0.767 | 0.759 | 0.763 | 0.469 | 0.400 |
>
> We also Validating Framework Effectiveness on Newer Base Models (Qwen3-Base-8B), Unlike the instruction-tuned variants used in our main experiments, Qwen3-8B only releases a base model without inherent instruction-following capabilities, so we only evaluate the fine-tuned model.  As shown in appendix B.1, the ReasonMap-Qwen3-8B model achieves better performance (F1: 0.860 on ID test set), outperforming both our previous Qwen2.5-based model and the proprietary API baselines.

---

> ### Author Response · Authors · 2025-11-25
>
> ### Response (Part 4)
>
> **Q7: Validate the downstream utility of MDTs using the generated solution (Weakness 5)**
>
>  We conducted two new sets of experiments.
> 1. Addressing "Marginal Improvement":  We hypothesized that the "marginal" gains observed previously might be due to the saturation of easier benchmarks. To test the true limit, we evaluated the highly challenging AIMO test Set (Olympiad-level https://huggingface.co/datasets/AI-MO/aimo-validation-aime) using a much stronger model, DeepSeek-V3.
>       - DeepSeek-V3: Accuracy improved significantly from 51.11% to 56.67%, a gain of +5.56%.
>       - Qwen2.5-7B-Instruct: Accuracy improved from 10.00% to 14.44%, a gain of +4.44%.
>
> | Model        | Base Accuracy | With MDTs | Improvement (Δ) |
> |--------------|---------------|-----------|-----------------|
> | Qwen2.5-7B   | 10.00%        | 14.44%    | +4.44%          |
> | DeepSeek-V3  | 51.11%        | 56.67%    | **+5.56%**      |
>
> 2. Inference via Self-Generated MDTs: To prove practical utility, we designed a "Self-Generated Hinting" pipeline that operates without any ground truth. We sample $k=4$ reasoning paths from the base model itself (Qwen2.5-7B-Instruct), use ReasonMap to extract MDTs from these potentially noisy rollouts, and feed the extracted MDTs back as hints for a final inference pass.
>  As shown in the table below (and detailed in Appendix D), this realistic setting consistently outperforms the base model.
>      - Math 500: Accuracy improves from 74.0% (Base) to 75.1% (Self-Generated MDTs).
>      - DeepMath-1K: Accuracy improves from 50.6% (Base) to 51.3% (Self-Generated MDTs).
>
>      It still yields gains (+1.1%), confirming that ReasonMap is not just "leaking" answers but effectively structuring the model's own knowledge to guide better reasoning.
>
> | Method Setting | Math 500 (Acc) | DeepMath-1K (Acc) |
> | :--- | :---: | :---: |
> | Qwen2.5-7B-Instruct (Base Model) | 0.740 | 0.506 |
> | **+ Self-Generated MDTs** | 0.751 | 0.513 |
> | *+ Oracle MDTs* | 0.759 | 0.518 |
>
> **Q8: Construction of DeepMath Test Set**
>
> The test set used in Section 4.5 is identical to the Out-of-Distribution (OOD) test set defined earlier in the paper.
> As detailed in Section 4.2 (Out-of-Distribution Evaluation), we constructed $\mathcal{D}\_{DeepMath}^{test}$ by uniformly sampling 1,000 instances from the large-scale DeepMath-103K dataset based on its domain distribution 1. This ensures representative coverage of diverse mathematical topics for our evaluation.
> The test set referenced in Section 4.5 for the downstream problem-solving task is exactly the same $\mathcal{D}\_{DeepMath}^{test}$. We used this set to evaluate whether MDTs could aid in solving highly complex problems that involve long-context reasoning.

---

> ### Comment · Reviewer_EYLJ · 2025-11-25
>
> I thank the reviewers for providing additional evidence, and for their thorough responses. I have raised my score from 2 to 6, as the majority of my concerns have now been addressed.

---

> > ### Author Response · Authors · 2025-11-26
> >
> > Thanks for raising the score! We would like to express our sincere gratitude to the reviewer for the thorough review and highly insightful comments. Those valuable suggestions have greatly helped us in improving the quality of our manuscript.
> >
> > Best regards.

---

### Official Review · Reviewer_dJsR · 2025-10-29

**Soundness:** 3
**Presentation:** 4
**Contribution:** 3
**Rating:** 6
**Confidence:** 3

**Summary:**

This paper introduces a new task, "Automatic Annotation of Mathematical Definitions and Theorems" (AAMDT), which aims to extract Mathematical Definitions and Theorems (MDTs) in a fine-grained manner from long Chain-of-Thought (CoT) reasoning processes. To address this task, the authors have designed a two-stage training framework named ReasonMap. The first stage, "Base Model Training," constructs a model with foundational knowledge and long-text generalization capabilities through Supervised Fine-Tuning (SFT) on both human-curated and LLM-augmented long CoT data. The second stage, "High-Fidelity Alignment," further refines the model using Direct Preference Optimization (DPO) to enhance the precision and reliability of the output. Experimental results show that ReasonMap significantly outperforms several strong baselines on the AAMDT task, especially in long CoT scenarios. Additionally, the authors validate the effectiveness of the extracted MDTs through a downstream math problem-solving task, demonstrating that they can serve as in-context prompts to enhance the reasoning abilities of other large language models.

**Strengths:**

1.  **Novel and Important Task:** The proposed AAMDT task addresses a key pain point in the current field of large model mathematical reasoning. Fine-grained knowledge annotation is crucial for understanding models' reasoning processes, conducting interpretability analysis, and constructing higher-quality training data. This work is highly pioneering and has significant potential for application.

2.  **Comprehensive Experiments:** The authors have conducted a series of thorough experiments to validate their method's effectiveness. Not only did they compare their method against multiple baselines on in-distribution (ID) and out-of-distribution (OOD) test sets, but they also demonstrated the necessity of each component in the framework (e.g., sequential training, definition-only samples) through detailed ablation studies. Furthermore, the downstream task validation directly demonstrates the practical value of the extracted MDTs.

**Weaknesses:**

1.  **Insufficient Representativeness of Models in Downstream Task Validation:** This is my primary concern. In Section 4.5, the authors use Qwen2.5-7B and Llama3.1-8B as base models to validate the improvement in problem-solving abilities from MDTs. Although these are excellent open-source models, they are not the current state-of-the-art Large Reasoning Models (LRMs). The performance gains observed on these models (around 1-2% on MATH500 and 1-2% on DeepMath), which have limited reasoning capabilities themselves, may not fully reflect the value of MDTs. It is possible that the reasoning capabilities of these models themselves are the bottleneck, limiting the potential impact of the MDTs. To more convincingly demonstrate the effectiveness of MDTs, I would be very eager to see experimental results on stronger, recognized expert models in mathematical reasoning. If the results are positive, I would be willing to raise my score.

2.  **Coarse-Grained OOD Evaluation Metric:** The authors use "domain relevance" as a proxy metric for OOD evaluation. While this is a viable compromise in the absence of fine-grained labels, its validity is debatable. This metric can only determine if an extracted MDT belongs to the broad domain of the problem but cannot measure its **effectiveness** or **non-triviality**. For example, in a calculus proof involving multiple complex theorems (like the Intermediate Value Theorem or L'Hôpital's Rule), if the model extracts overly basic and broad labels such as "Definition: Set" or "Definition: Real Number," it would be considered correct under the current OOD evaluation because it is indeed related to the domain of mathematics. However, such "correct but uninformative" labels are of little help in understanding the proof or aiding reasoning, and may even introduce noise due to their redundancy (and this situation is indeed prevalent in your publicly released dataset).

3.  **Potential Misalignment in Long CoT Generation:** In constructing the D_long dataset, the authors instruct the model to "pretend it has not seen the original proof" and generate a new long CoT based on a brief human-written proof outline. This process presents a subtle challenge. Based on my experience and testing, the model is likely to fall into one of two extremes: a) **Path Deviation:** The model fails to strictly follow the core logic of the original proof, introducing new theorems or steps not present in the original. This creates a factual misalignment between the generated long CoT (A_l) and the original MDT labels (L), resulting in noisy data. b) **Over-Imitation:** The model simply expands and "colloquializes" each step of the short proof without genuinely exhibiting a complex, exploratory CoT process. This undermines the original purpose of using this data for "long CoT generalization training." While this does not significantly undermine the paper's persuasiveness, I hope the authors can elaborate more on the limitations of this process.

**Questions:**

1.  Regarding the downstream task validation, have you considered conducting experiments on stronger reasoning models (e.g., GPT-4o, DeepSeek-V3)? Are the performance gains from MDTs still significant on these models? This is crucial for assessing the true potential of this work.

2.  Regarding the OOD evaluation metric, are the authors aware of the potential issues caused by its coarse granularity (i.e., rewarding overly broad or trivial MDTs)? Could you comment on the frequency of this situation in your tests? Have you considered supplementary evaluation methods, such as manual evaluation of a small sample to measure the **informativeness** or **utility** of the extracted MDTs?

3.  When generating the long-chain reasoning data (D_long), how did you ensure that the generated content remained logically consistent with the original proof outline while avoiding simple paraphrasing, thereby ensuring the data's high quality and alignment? Was there any data cleaning or filtering process involved?

---

> ### Author Response · Authors · 2025-11-25
>
> We sincerely thank the reviewers for their insightful comments and constructive feedback. In response to your concerns, we have conducted additional experiments and analyses. Our detailed responses are as follows:
>
> **Q1: Insufficient Representativeness of Models in Downstream Task Validation (Weakness 1)**
>
> To demonstrate the effectiveness of MDTs on more powerful Large Reasoning Models, we conducted a new experiment using DeepSeek-V3 on the highly challenging AIMO testset (Math Olympiad level https://huggingface.co/datasets/AI-MO/aimo-validation-aime).
>
> | Model        | Base Accuracy | With MDTs | Improvement (Δ) |
> |--------------|---------------|-----------|-----------------|
> | Qwen2.5-7B   | 10.00%        | 14.44%    | +4.44%          |
> | DeepSeek-V3  | 51.11%        | 56.67%    | **+5.56%**      |
>
> As shown in the newly added Appendix B.2 (Table 6), explicitly injecting MDTs extracted by ReasonMap yielded significant performance gains:
> - DeepSeek-V3: Accuracy improved from 51.11% to 56.67%, a remarkable gain of +5.56%.
> - Qwen2.5-7B: Accuracy improved from 10.00% to 14.44% (+4.44%).
>
> Experimental results show that accurate MDTs can offer a more significant increase on stronger models in mathematical reasoning
>
> **Revision**: We added this content to Appendix B.2
>
> **Q2: OOD Metric Validity and Triviality Analysis (Weakness 2)**
>
> We agree there is a risk that it could reward trivial tags (e.g., "Definition: Real Number") that are technically correct but uninformative. To address this, we conducted a Human Triviality evaluation and a deeper analysis of our training data structure.
> Following your suggestion, we quantified the prevalence of "correct but uninformative" tags in our OOD results. We randomly sampled 300 extracted MDTs that were judged as "domain-relevant" by the LLM. Three mathematics PhDs evaluated these to determine if they were "trivial" (i.e., generic terms providing no specific reasoning value). As detailed in Appendix D.3, human experts classified only 18.3% of the valid MDTs as trivial.
>
> **P.s.**
>
> The presence of basic definitions is a necessary consequence of our rigorous data source, ProofWiki (via NaturalProofs-Gen). ProofWiki mandates explicit annotation even for basic axioms (e.g., Definition: Real Number) to ensure that every logical step is traceable and formally complete (e.g. https://proofwiki.org/wiki/Combination_Theorem_for_Sequences/Complex/Product_Rule).
> Our ablation study (dropping 1k/3k definitions) showed consistent performance drops in both ID and OOD evaluations. This proves that these "basic" definitions are not noise. And naturalprover also include definition loss to train the prover model.
>
> **Q3: Quality Assurance of Synthetic Long CoTs (Weakness 3)**
>
> We address each concern with quantitative evidence and experimental validation. To verify that the model is engaging in genuine "Reasoning Stylization" rather than simply "paraphrasing" the human proof, we conducted a ROUGE analysis between the source concise proofs ($A$) and the generated Long CoTs ($A_l$)
>
> | Metric   | Precision | Recall | F1-Score |
> |----------|-----------|--------|----------|
> | ROUGE-1  | 0.1089    | 0.7398 | 0.1771   |
> | ROUGE-2  | 0.0578    | 0.4135 | 0.0948   |
> | ROUGE-L  | 0.0776    | 0.5720 | 0.1280   |
>
> In the "w/o hint" experiment, we removed the constraint, allowing the model to solve the problem freely. This frequently resulted in the model generating a new solution path $A'$ (deviating from the human solution). However, since the Ground Truth $L$ was still tied to the original path $A$, this created a mismatch (Label Noise). The results showed a decline in performance.
>
> These results show that the augmentation faithfully diversifies the reasoning data and maintains the correct answer, instead of introducing noise and bias.
>
> **Revision**: We added this content to Appendix E.

---

> ### Author Response · Authors · 2025-11-28
>
> As the discussion phase is approaching its end, we kindly request the reviewer to let us know if our response has addressed your concerns, particularly regarding the validation on stronger models (where we achieved a +5.56% gain on DeepSeek-V3).
>
> We hope the clarifications and the added experiments resolve the remaining questions. We would be happy to address any additional points the reviewer may have during the remaining time of the discussion phase. We thank the reviewer for engaging with us in the discussion.
>
> Best regards, Authors

---

### Official Review · Reviewer_DW6j · 2025-11-01

**Soundness:** 3
**Presentation:** 3
**Contribution:** 3
**Rating:** 2
**Confidence:** 4

**Summary:**

The paper introduces the Automatic Annotation of Mathematical Definitions and Theorems (AAMDT) task, aimed at identifying fine-grained mathematical definitions and theorems within large language models’ (LLMs) chain-of-thought (CoT) reasoning. To tackle this, the authors propose ReasonMap, a two-stage framework. In Stage 1 (Foundational Model Training), a hybrid dataset combining concise human-written proofs and long LLM-augmented CoTs is used to teach both accuracy and long-context understanding. Stage 2 (High-Fidelity Alignment) refines this model using Direct Preference Optimization (DPO) to reduce hallucinations and formatting inconsistencies. Experimental results across in-distribution and out-of-distribution benchmarks show that ReasonMap significantly outperforms strong baselines such as GPT-4o-mini and DeepSeek-V3, especially for complex reasoning tasks. The extracted annotations (Mathematical Definitions and Theorems, MDTs) also improve the performance of other reasoning models when injected as contextual hints. This work provides a scalable, cost-effective way to generate high-quality, fine-grained mathematical annotations and enhances interpretability and downstream reasoning quality in LLMs.

**Strengths:**

1. The novel task definition (AAMDT) fills a crucial gap in mathematical reasoning datasets.
2. Two-stage framework effectively combines supervised fine-tuning and preference alignment.
3. Empirical validation is thorough—covering in-distribution, out-of-distribution, and downstream reasoning impact. Clear scalability advantage, reducing reliance on manual annotation.

**Weaknesses:**

1. The paper claims to annotate `fine-grained` Mathematical Definitions and Theorems (MDTs) but does not clearly formalize what level of granularity qualifies as fine-grained. Without a standardized taxonomy or ontology (e.g., linking MDTs to known mathematical knowledge graphs), reproducibility and consistency across datasets remain unclear.
2. The long CoT augmentation relies on DeepSeek-R1 generations. This introduces an uncontrolled source of noise and style bias. No explicit data filtering or quality assurance metrics are reported for these synthetic CoTs.
3. The baselines (GPT-4o-mini, DeepSeek-V3, Qwen, Llama) are used in few-shot prompting setups, not fully fine-tuned under the same conditions. This weakens the claim that ReasonMap significantly outperforms other methods. I think the improvements could stem from different tuning or dataset exposure rather than intrinsic superiority.
4. The F1 and accuracy metrics depend on another LLM’s judgement matching of MDTs. Any human evaluation or inter-annotator agreement study could be provided?
5. The rejected examples in DPO training are generated by the same model $\pi$ long, filtered only by a scalar threshold. This can lead to homogeneous (i.e., the bad examples tend to look very similar to the good ones), limiting the robustness of alignment and making the model overfit to its own prior mistakes. If the model only learns from its own style of mistakes, it won’t see enough diverse or surprising wrong answers.

**Questions:**

As my understanding, the paper says "it annotates mathematical definitions and theorems (MDTs) used in reasoning chains", it assumes that each problem has one correct set of theorems and definitions used in the solution (ground truth). But in mathematics, there are often multiple valid ways to solve the same problem — you might use a different theorem or definition to reach the same correct answer. So when ReasonMap compares its generated annotations against that single “ground truth” list, it might mark an answer as wrong just because it used a different but still correct theorem.

---

> ### Author Response · Authors · 2025-11-25
>
> ### Response (part 1)
>  We sincerely thank the reviewers for their time and valuable feedback on our work. Below, we provide detailed responses to the questions and concerns raised.
>
> **Q1: Concern about mismarking answers and how we handle Multiple Valid Solutions**
>
> ReasonMap does not penalize valid alternative solutions because our Ground Truth is not defined at the "Problem ($q$)" level, but rigorously bound to the "Reasoning Chain/Solution ($A$)" level. We assume that a single problem can have multiple solutions, and we have handled that properly. Specifically, each training/test instance is a triplet $q, A, L$, where $L$ (MDTs) specifically annotates the logic used in solution $A$.  If Problem $Q$ has two valid solutions—Solution $A$ (using Geometry Axioms) and Solution $B$ (using Algebraic Coordinates)—they are treated as two distinct instances in our evaluation: $\langle Q, A, L_A \rangle$ and $\langle Q, B, L_B \rangle$.
>
> **Evidence**:
> Our ablation study in Section 4.4 ("w/o hint") 1 empirically validates this design choice by simulating exactly the "misalignment" scenario the reviewer is concerned about. In the "w/o hint" experiment, we removed the constraint, allowing the model to solve the problem freely. This frequently resulted in the model generating a new solution path $S'$ (mathematically correct but different). However, since the Ground Truth $L$ was still tied to the original path $S$, this created a mismatch (Label Noise). The results showed a decline in performance.
>
> **W1: Formalizing Granularity (what is fine-grained)**
>
> Our definition is relative to existing work and explicitly states the source of our taxonomy to ensure reproducibility. We define the granularity levels as follows, which is now explicitly detailed in the revised caption of Figure:
> - Coarse-grained: Refers to high-level domain tags (e.g., "Algebra", "Geometry") used in previous datasets.
> - Fine-grained: Refers to specific Atomic Knowledge Units (e.g., "Principle of Mathematical Induction", "Definition: Cyclic Group").
>
> Our annotations are not arbitrarily created. They are derived directly from ProofWiki, a structured, expert-curated mathematical knowledge base.  We ensure that our "fine-grained" labels are standardized, consistent, and reproducible across different datasets. This links our extracted MDTs to known mathematical knowledge, addressing the concern about external validity. Here is an original example: https://proofwiki.org/wiki/Combination_Theorem_for_Sequences/Complex/Product_Rule.
>
> **W2:  Missing quality assurance metrics  for these synthetic CoTs.**
>
> The synthetic process is strictly a controlled augmentation, instead of uncontrolled generation. We utilize the correct human answer ($A$) as a ground-truth guidance, prompting DeepSeek-R1 to "elaborate" rather than solve from scratch. We also manually verify that the augmented solutions are correct. This strategy intentionally introduces the verbose "style bias" characteristic of modern LLMs to ensure realistic distribution coverage, while strictly preserving the rigorous logic of the original human proofs.
> We validate the quality and alignment of this augmentation through two dimensions:
> 1. Implicit Assurance (Ablation Study): As shown in Sec 4.4 ("w/o hint"), removing the human answer constraint leads to a significant performance drop. This confirms that the "guidance" is essential for preventing logical drift.
> 2. Quantitative Metric (ROUGE Analysis): A comparison between source proofs ($A$) and generated CoTs ($A_l$) reveals high recall (0.74) but low precision (0.11). This specific metric profile quantitatively confirms our objective: The augmentation faithfully diversifies the reasoning data and maintains the correct answer, instead of introducing noise and bias.
>
> | Metric   | Precision | Recall | F1-Score |
> |----------|-----------|--------|----------|
> | ROUGE-1  | 0.1089    | 0.7398 | 0.1771   |
> | ROUGE-2  | 0.0578    | 0.4135 | 0.0948   |
> | ROUGE-L  | 0.0776    | 0.5720 | 0.1280   |
>
> **Revision**: We have added the evaluation result in Appendix D.1

---

> ### Author Response · Authors · 2025-11-25
>
> ### Response (Part 2)
>
> **W3: Fairness of Baselines (Fine-tuned vs. Few-shot)**
>
> Our objective is not to claim architectural superiority over frontier models like GPT-4o-mini, but to demonstrate the practical viability of a specialized, cost-effective annotator.
> AAMDT is a fine-grained extraction task requiring strict format fidelity. Generalist LLMs, despite their power, often struggle to achieve the necessary precision solely through few-shot prompting. Consequently, current large-scale annotation efforts (e.g., DeepMath-103K, Omni-Math) rely on prohibitively expensive commercial APIs. We demonstrate that a compact open-source model—optimized via our "one-time" SFT+DPO framework—serves as a scalable, high-performance alternative to these costly APIs by including an Approximate cost comparison between the GPT5-mini API and our ReasonMap-7B model (see appendix F)
> Besides, comparing specialized fine-tuned models against generalist closed-source APIs is a standard practice in recent literature when proposing task-specific solutions:
> - LIMO compares fine-tuned Qwen2.5-32B against OpenAI-o1-preview[1].
> - SuperCorrect [2]and ReasonFlux [3] compare fine-tuned 7B models against GPT-4o to demonstrate domain-specific efficacy.
>
> [1] Ye, Yixin, et al. "Limo: Less is more for reasoning." arXiv preprint arXiv:2502.03387 (2025).
>
> [2] Yang, Ling, et al. "SuperCorrect: Advancing Small LLM Reasoning with Thought Template Distillation and Self-Correction." The Thirteenth International Conference on Learning Representations.
>
> [3] Yang, Ling, et al. "Reasonflux: Hierarchical llm reasoning via scaling thought templates." arXiv preprint arXiv:2502.06772 (2025).
>
> **W4: Lack of human evaluation to further validate LLM’s judgment**
>
> We conducted a human evaluation of the LLM-Judge (GPT-4o-mini) results.
> We randomly sampled 100 evaluation instances (50 pairs for Fine-grained MDT matching from the ID set and 50 pairs for Domain Relevance from the OOD set). Three independent human experts (PhDs in Mathematics) match the model output with ground truth to determine semantic equivalence. We calculated the Precision and Recall of the LLM-Judge's results against the human experts' results.
>
> | Evaluation Task | Human Expert | Precision | Recall |
> |-----------------|--------------|-----------|--------|
> | **Domain Relevance** | Annotator A | 0.9083 | 0.9033 |
> | | Annotator B | 0.9100 | 0.9070 |
> | | Annotator C | 0.9050 | 0.9080 |
> | | *Average* | *0.9078* | *0.9061* |
> | **Fine-grained MDTs** | Annotator A | 0.9240 | 0.9330 |
> | | Annotator B | 0.9190 | 0.9280 |
> | | Annotator C | 0.9300 | 0.9260 |
> | | *Average* | *0.9243* | *0.9290* |
>
> The high agreement rate ($>90\%$) between human experts and the LLM judge confirms that our reported F1 and Accuracy metrics reflect genuine semantic extraction.
>
> **Revision**: We added this content to D.1.
>
> **W5: Concern about Homogeneous Negatives in DPO stage**
>
> We respectfully offer a different perspective supported by our empirical findings: for fine-grained extraction tasks, homogeneity (similarity between Chosen and Rejected) is a feature, not a bug.
> Inspired by prior work [1], we intentionally target "Hard Negatives"—candidates that are semantically close to the ground truth but contain subtle hallucinations.  By strictly filtering with a high scalar threshold from the model's own distribution, we force the model to distinguish between correct logic and plausible but flawed hallucinations.
> Our method is directly supported by the ablation study on the DPO quality threshold ($\gamma$) in Section 4.4 (Table 2):
> - $\gamma=0.9$ (Hard/Homogeneous): Selecting negatives with high similarity to Ground Truth consistently yielded superior performance.
> - $\gamma=0.5$ (Easy/Diverse): Selecting "easier" or more diverse negatives resulted in suboptimal alignment.
>
> This confirms that for the AAMDT task, the model benefits more from correcting its own hard negative cases than from seeing diverse but easily distinguishable errors.
>
> [1] Cheng, Jiale, et al. "SPaR: Self-Play with Tree-Search Refinement to Improve Instruction-Following in Large Language Models." The Thirteenth International Conference on Learning Representations.
>
> **Revision**: We updated the description in Section 3.2.2, confirming our motivation.

---

> ### Author Response · Authors · 2025-11-28
>
> As the discussion phase is approaching its end, we hope the above clarifications and the additional experiments in the revised draft sufficiently addressed your concerns. If you are satisfied, we kindly request you to consider updating the score to reflect the newly added results and discussion. We remain committed to addressing any remaining points you may have during the discussion phase.
>
> Best regards, Authors

---

### Official Review · Reviewer_ykxB · 2025-11-03

**Soundness:** 2
**Presentation:** 3
**Contribution:** 2
**Rating:** 4
**Confidence:** 3

**Summary:**

ReasonMap approaches an elicitation-based (‘fine-grained’) extraction of mathematical definitions and theorems (MDTs) from long chain-of-thought solutions via a two-stage pipeline: foundational SFT on more concise proofs plus LLM-augmented short to long CoTs, followed by DPO with negatives selection. On long-CoT tests it outperforms baselines and shows OOD gains. The proposed solution has practical implications in the area of Mathematical Reasoning, but there are some questions wrt the depth of analysis and communication of the contribution (articulated below). The proposed approach looks promising: addressing some of the limitations may substantially improve the long term impact.

**Strengths:**

- Clearly articulated/described two-stage training.

- Relevant empirical results with mid-size models beat larger baselines on long-CoT extraction by notable F1 margins. Gains persist on OOD sets.

- Overall downstream utility and applicability.

**Weaknesses:**

- Lack of a better articulation of the underlying principle.

- The paper lacks a clear structural model on the assumptions behind mathematical reasoning.

- Variability in reasoning structure across datasets (styles, domains) is neither analyzed nor controlled, limiting external validity

- Baseline selection is opaque (inclusion/exclusion criteria).

- Coverage of related work is incomplete and does not adequately position the contribution relative to math (and more general) CoT supervision. One example: https://arxiv.org/pdf/2502.12616

- Lack of more extensive qualitative analyses.

**Questions:**

* Qualitative analysis/Interpretation.

- What are the dominant false-positive/false-negative patterns (e.g., confusing lemmas with definitions, inference drift, spurious generic statements)?

- How do errors differ across your proposed solution and the baselines? What are the side effects?

* Experimental design.

- What is the formal selection (inclusion and exclusion) for the baselines? Can you elicit them more formally?

- How does the system handle adversarial or misleading CoTs (negations, swapped order, aliasing of named theorems, step omissions, LaTeX variants, informal paraphrases)?

- How is the variability within LLM family and out-family?

Mechanism.

- What are the underlying assumptions behind your selected datasets (these are your proxies for mathematical reasoning)?

- Can you state the key underlying principles of the underlying mechanism that the proposed approach aims to achieve? What is being hypothesized as a representational effect (apart from the task scores)?

---

> ### Author Response · Authors · 2025-11-25
>
> ### Response (part1)
> Thanks for the valuable feedback and helpful suggestions. We will add more explanation to the questions. Thanks for pointing out the weaknesses and problems. Here are our explanations:
>
> **Q1.1: Qualitative analysis/Interpretation on Dominant Error Patterns (Weakness 2.2)**
>
> We conducted a human error analysis evaluation involving three PhDs in Mathematics on 100 failure cases (defined as instances where labels overlap $<$ 50%) from both the baseline (DeepSeek-V3) and our model (ReasonMap-7B). Human evaluators classified errors into three distinct categories based on the nature of the misalignment:
> - Code A (Granularity Mismatch): The predicted concept is semantically correct but differs in specificity (e.g., predicting "Mean Value Theorem" instead of "Cauchy's Mean Value Theorem"). These are considered "recoverable" errors.
> - Code B (Implicit Omission): The model fails to extract a critical, non-trivial MDT present in the ground truth. This represents a "conservative" failure mode.
> - Code C (Inference Drift / Hallucination): The model predicts MDTs that are clearly irrelevant or semantically distant from the ground truth. These are the most destructive errors as they introduce noise and misinformation.
>
> **Q1.2 Errors difference between the baseline and our method, and the side effect (Weakness 2.2)**
>
> We told the human evaluators to classify 100 failure cases from both DeepSeek-V3 and our model. The distribution of these errors is presented in the table below:
> | Model        | Code A (Granularity) | Code B (Omission) | Code C (Drift) |
> |--------------|---------------------|------------------|----------------|
> | DeepSeek-V3  | 10%                 | 32%              | **58%**        |
> | ReasonMap    | 12%                 | 40%              | **48%**        |
>
> The result reveals a significant shift in error distribution between the baseline and our proposed solution:
> - Suppression of Inference Drift： DeepSeek-V3’s dominant failure mode is Code C (58%), indicating a tendency to "hallucinate" connections or retrieve irrelevant concepts. ReasonMap successfully reduces this high-risk category by 10 percentage points (to 48%). This demonstrates that our training strategy (SFT+DPO) effectively suppresses the generation of noisy and unreliable concepts.
> - Side Effects (The Trade-off): We explicitly acknowledge a side effect of our alignment strategy: an increase in Code B (Implicit Omission) (40% vs. 32%). This suggests that ReasonMap adopts a more conservative strategy. By suppressing excessive output to reduce hallucinations (Code C), the model effectively raises its internal confidence threshold, which inevitably leads to missing some valid but less obvious key information.
>
> **Revision**: We added this content to Appendix D.2
>
> **Q2.1: Formal Selection Criteria for Baselines**
>
> We select frontier general-purpose instruction-following models (both open and closed source) that represent the current SOTA level in terms of cost-performance ratio. And we include GPT-4o-mini and DeepSeek-V3. Acknowledging the rapid iteration of these model families, we have expanded our evaluation to include their latest versions, DeepSeek-V3.1 and GPT-5-mini.
>
> **Revision**: We update this content in Section 4.2
>
> **Q2.2: How does the system handle adversarial or misleading CoTs? （Weakness 1.2）**
>
> ReasonMap functions as a Mathematical Knowledge Extractor, distinct from a Mathematical Proof Verifier.
> To bolster robustness against stylistic variations—such as informal paraphrases, LaTeX formatting differences, and theorem aliasing—we transform concise human-written proofs into noisy, verbose LLM-generated contexts via generative restyling. To prevent unconstrained hallucination and ensure alignment with ground-truth MDTs, we employ Guided Augmentation. Specifically, the CoT generation is explicitly conditioned on and guided by the correct human solution. This methodology aligns with recent findings from REER [1], guiding intermediate CoT construction with provided ground-truth answers.
>
> [1] Wang, Haozhe, et al. "Reverse-engineered reasoning for open-ended generation." arXiv preprint arXiv:2509.06160 (2025).
>
> **Q2.3: The variability within LLM family and out-family （Weakness 1.2）**
>
> We select the open-source base models that serve as the backbones for ReasonMap. We focus on widely adopted dense models, specifically the Llama and Qwen series. We implement ReasonMap using the instruct versions of Qwen2.5-7B and Llama3.1-8B to investigate out-of-family performance across different architectures. Additionally, we include the latest Qwen3-8B (Base) model to enable a within-family comparison.
>
> **Revision**: We have updated the evaluation result of Qwen3-8B Base in Appendix B.1

---

> ### Author Response · Authors · 2025-11-25
>
> ### Response (part 2)
>
> **Q3.1 Assumptions behind our selected datasets  (Weakness 1)**
>
> Our work is built upon two fundamental hypotheses regarding mathematical cognition:
> - Decomposability: We posit that complex mathematical reasoning is not merely continuous text generation but can be fundamentally decomposed into a series of discrete reasoning steps anchored by Atomic Knowledge Units (MDTs). This assumption is supported by the structure of our source data, NaturalProofs-Gen, where proofs are explicitly constructed as sequences of steps linked to specific MDTs.
> - Path-Knowledge mapping: Prior work like NaturalProver [1] demonstrated that conditioning on MDTs can help generate valid reasoning paths (Knowledge $\to$ Path). Our AAMDT task proposes the inverse problem: given a complex CoT path, can we reliably extract the unique set of supporting MDTs (Path $\to$ Knowledge)? ReasonMap is fundamentally designed to learn this reverse mapping.
>
> So to achieve robust Path-Knowledge mapping, we include 3 training dataset,  First, $\mathcal{D}_{acq}$,  we  utilize it to provide a strong supervision signal linking concise steps to MDTs. This teaches the model the fundamental logic of which knowledge points ($L$) are derived from which reasoning steps ($A$).
>
> The second dataset is $\mathcal{D}\_{long}$.
> We hypothesize that $\mathcal{D}\_{acq}$
> teaches the model the semantics of concepts (e.g., what "Mathematical Induction" is), while $\mathcal{D}_{long}$ teaches the model how to locate and apply these concepts within noisy and verbose LLM contexts. This necessitates our "Sequential Training" curriculum (learning definitions first, then application), the superiority of which is empirically confirmed by our ablation study.
> The last dataset is DPO dataset: we teach the model to learns to distinguish between the Ground Truth (Chosen) and "plausible but slightly flawed" answers through DPO.
>
> [1] Welleck, Sean, et al. "Naturalprover: Grounded mathematical proof generation with language models." Advances in Neural Information Processing Systems 35 (2022): 4913-4927.
>
> **Revision**: We have updated this content to Section 3.1
>
> **Q3.2: Underlying Principles, Assumptions, and Mechanisms (Weakness 1)**
>
> We acknowledge that our initial manuscript prioritized the methodological implementation of theoretical principles.
> Our proposed framework operates in two distinct stages. In Stage 1, our primary goal is to obtain a foundational model with robust MDT extraction capabilities. We initially train on  $\mathcal{D}\_{acq}$
>   to establish this; however, since $\mathcal{D}\_{acq}$ consists exclusively of concise human solutions, the resulting model exhibits a propensity for  low generability on CoT, particularly when processing verbose, long-form Chain-of-Thought (CoT) contexts. Consequently, we implement a long-CoT generalization phase within Stage 1, which significantly mitigates this issue, as evidenced by the narrowed gap between recall and precision in our results table. Finally, in Stage 2, we introduce DPO to force the model to distinguish Ground Truth from "plausible but flawed" hallucinations, further refining the model's adaptability to complex, long-context scenarios.
>
> **Weakness 2.1: Coverage of Related Work and Positioning**
>
> We appreciate the reviewer for directing our attention to the relevant work. We have carefully reviewed this paper. We would like to clarify the distinct positioning of our research compared to this work:
> - The cited work primarily focuses on "Supervised Chain-of-Thought (CoT) Generation"—aiming to teach models how to construct better reasoning paths from scratch. In contrast, ReasonMap operates as a "Knowledge Extractor." Our core objective is not to generate the reasoning itself, but to analyze existing reasoning chains
> - Our work is highly complementary to CoT supervision frameworks (in arXiv:2502.12616). The fine-grained MDT knowledge units extracted by ReasonMap can serve as high-quality supervision signals or reward functions for training frameworks.

---

> ### Author Response · Authors · 2025-11-28
>
> As the discussion phase is approaching its end, we would like to briefly highlight how our revision addresses your key concerns regarding the mechanism and qualitative understanding of our method:
>
> Underlying Principles (Weakness 1): We have explicitly articulated the "Decomposability" and "Path-Knowledge Mapping" hypotheses in the revised draft to clarify the theoretical foundation of ReasonMap.
>
> Qualitative & Error Analysis (Weakness 2 & Questions): We conducted a detailed human evaluation (with math experts) on failure cases. The results reveal that our method significantly suppresses harmful "Inference Drift" (Hallucination) compared to the baseline (reducing it from 58% to 48%), essentially trading off some recall for higher reliability—a trade-off we now explicitly discuss.
>
> We hope the above clarifications and the additional experiments in the revised draft sufficiently addressed your concerns. If you are satisfied, we kindly request you to consider updating the score to reflect the newly added results and discussion. We remain committed to addressing any remaining points you may have during the discussion phase.
>
> Best regards, Authors

---

### Author Response · Authors · 2025-12-02
**Response Summary**

We sincerely appreciate the Area Chair's dedication and the reviewers' constructive feedback. We are pleased to report that Reviewer EYLJ has raised their score from 2 to 6 . Notably, Reviewer EYLJ's thorough reviews (5 weaknesses, 9 questions) encompassed the primary concerns shared by other reviewers; thus, this score increase serves as strong validation that these overlapping issues have been effectively resolved.

| **Reviewer** | **Original → Updated Score** | **Discussion Status** | **Key Acknowledgements / Remaining Concerns**                |
| ------------ | ---------------------------- | --------------------- | ------------------------------------------------------------ |
| **EYLJ**     | **2 $\to$ 6**                | Done                  | 1. Stated "majority of my concerns have now been addressed"  |
| **dJsR**     | 6                            | not start             | 1. Explicitly stated willingness to raise score if results on stronger models were positive; 2. We have **met this condition** by demonstrating significant gains (+5.56%) on DeepSeek-V3. |
| **ykxB**     | 4                            | not start             | No discussion due to system issue.                           |
| **DW6j**     | 2                            | not start             | No discussion due to system issue.                           |

## Recognized Strengths

- **Pioneering Task & Practical Potential:** The proposed AAMDT task addresses a critical gap in mathematical reasoning by enabling fine-grained knowledge annotation, with significant potential for interpretability and training data construction (dJsR, DW6j). Reviewer ykxB explicitly acknowledges the approach as **"promising,"** highlighting its substantial potential for long-term impact (ykxB).
- **Effective Two-Stage Framework:** The ReasonMap pipeline (SFT + DPO) is clearly articulated and effectively combines foundational learning with high-fidelity alignment (ykxB, DW6j, EYLJ).
- **Strong Empirical Performance:** The method consistently outperforms baselines on both ID and OOD datasets, with ablation studies validating the necessity of each component (ykxB, dJsR, EYLJ).
- **Scalability & Reproducibility:** The work offers a cost-effective, scalable alternative to manual annotation and proprietary APIs, with all materials provided for reproducibility (DW6j, EYLJ).

## Addressed Concerns

- **Evaluation on Frontier Models (EYLJ W4, Q6):** We extended our evaluation to include  DeepSeek-V3.1, and GPT-5-mini. Results confirm that ReasonMap-7B continues to achieve superior fine-grained extraction (e.g., F1 **0.775** vs. DeepSeek-V3 F1 **0.735** on $\mathcal{D}_{acq}^{test}$) and validated our framework on a newer base model (Qwen3-8B), achieving an F1 of **0.860**, further proving the framework's adaptability.
- **Theoretical Motivation & Data Quality (EYLJ W1; ykxB W1-3, Q3; dJsR W3, Q3; DW6j W2-3):** Revised Sections 2-3 to incorporate more related references, clarify theoretical motivation,  and clarified the hypothesis & principles, verified the quality of synthetic long-CoT data via ROUGE analysis.
- **Downstream Utility & Realism (EYLJ W5, Q7; dJsR W1, Q1):**
  - To address questions  about "marginal" gains, we evaluated on the challenging AIMO benchmark. Injecting MDTs improved DeepSeek-V3 performance by **+5.56%** (51.11% $\to$ 56.67%). This positive result directly satisfies the condition set by Reviewer dJsR.
  - To address questions about "unrealistic" settings, we use ReasonMap extracts MDTs from the model's own noisy rollouts. This yielded consistent gains on Math500 and DeepMath-1K.
- **Human evaluation (EYLJ W3, Q2-3; dJsR W2, Q2; ykxB Q1; DW6j W4):**
  - Metric Validity: We validated GPT-4o-mini's judgment against human experts, achieving **>90%** agreement (Precision 0.924, Recall 0.933), implemented human evaluation revealed that only **18.3%** of extracted MDTs are "trivial" (In our OOD evaluation).
  - Error Analysis: We analyzed failure cases, showing that ReasonMap significantly suppresses "Inference Drift" (hallucinations) compared to baselines (**48% vs. 58%**).
- **Clarification on Multiple Solutions (DW6j Q1):** We corrected a fundamental misunderstanding regarding how ReasonMap handles multiple valid solutions. We clarified that our Ground Truth is rigorously bound to the specific Reasoning Chain ($A$), not the Problem ($Q$).  Our ablation study confirmed that decoupling the solution from the annotation leads to label noise and performance degradation.
- **Cost-Effectiveness Analysis (EYLJ W2, Q1):** We provided a detailed cost comparison for processing 1M samples. The ReasonMap ($\approx$ **$484**) v.s.  GPT-5-mini API ($\approx$ **$1,500**).

We have updated the draft to reflect these experimental results and clarifications.

---

### Meta-Review · Area_Chair_8Z1x · 2026-01-09

**Summary:**

The paper proposes a definition/theorem way to structuring the CoT process such that the reasoning process can be better navigated towards the final answer. Then the paper uses conventional alignment techniques to finetune the LLMs. The results seem pretty decent.

**Reviewer Concerns:**

- The major concern is the lack of underlying principle for the proposed technique. It looks more like an interesting trick. After reading the paper carefully, I agree that this is a valid concern. I also found no clear principles behind the proposed method

- A lot of concerns are for the baseline methods. It seems that the paper didn't use the strongest and most fair baselines. I think the authors spent a lof of efforts to address this. I can sense the amount of efforts is huge. Thus I think the concerns from the empirical side is partially addressed.

- Moreover, some reviewer think the performance improvement is quite marginal, which I also agree.

Combining all the information, I think the paper is still below the bar of acceptance.

**Reviewer Scores:**

I think the final score for four reviewers are 6,6,4,2 after the rebuttal. One reviewer changed from 2 to 6.

---

### Decision · Program_Chairs · 2026-01-26

Reject